# Large-scale orientational order in bacterial colonies during inward growth

**Mustafa Basaran[1,2†], Y Ilker Yaman[1†], Tevfik Can Yüce[1], Roman Vetter[3‡], Askin Kocabas[1,2,4,5]***

[1]Department of Physics, Koç University, Istanbul, Turkey; [2]Bio-Medical Sciences and Engineering Program, Koç University, Istanbul, Turkey; [3]Computational Physics for Engineering Materials, ETH Zurich, Zurich, Switzerland; [4]Koç University Surface Science and Technology Center, Koç University, Istanbul, Turkey; [5]Koç University Research Center for Translational Medicine, Koç University, Istanbul, Turkey

**\*For correspondence:**
akocabas@ku.edu.tr

[†]These authors contributed equally to this work

**Present address:** [‡]Department of Biosystems Science and Engineering, ETH Zurich, Basel, Switzerland

**Competing interest:** The authors declare that no competing interests exist.

**Abstract** During colony growth, complex interactions regulate the bacterial orientation, leading to the formation of large-scale ordered structures, including topological defects, microdomains, and branches. These structures may benefit bacterial strains, providing invasive advantages during colonization. Active matter dynamics of growing colonies drives the emergence of these ordered structures. However, additional biomechanical factors also play a significant role during this process. Here, we show that the velocity profile of growing colonies creates strong radial orientation during inward growth when crowded populations invade a closed area. During this process, growth geometry sets virtual confinement and dictates the velocity profile. Herein, flow-induced alignment and torque balance on the rod-shaped bacteria result in a new stable orientational equilibrium in the radial direction. Our analysis revealed that the dynamics of these radially oriented structures, also known as aster defects, depend on bacterial length and can promote the survival of the longest bacteria around localized nutritional hotspots. The present results indicate a new mechanism underlying structural order and provide mechanistic insights into the dynamics of bacterial growth on complex surfaces.

## Editor's evaluation

The growth of bacterial colonies on solid substrates is a common assay used in a variety of settings, from probing bacterial organization in biofilms to spatial population genetics. The common setup is an outward growing colony from a central seed. In this work, Basaran et al., study a colony growing inward from an annulus. The authors show that this geometrical modification has profound consequences on the alignment of rod-shaped bacteria. This is caused by a flow alignment effect, and lead to a radial ordering reminiscent of an aster or +1 topological defect. This result is motivated by experimental observations with *E. coli* and interpreted using modern active matter theories, with ample support from extensive numerical simulations of detailed finite element and continuum models.

## Introduction

Bacterial colonization and invasion are collective phenomena. These processes are regulated through a complex interplay of physical and biological interactions in a crowded population. Bacterial morphology, hydrodynamics, surface topology, and topography markedly alter growth mechanisms, morphology, and overall competition among bacteria (*Grant et al., 2014*; *Su et al., 2012*; *Volfson et al., 2008*; *Warren et al., 2019*; *Cho et al., 2007*; *Smith et al., 2017*). Elucidation of the factors

regulating collective bacterial growth and their competition is essential to enhance our understanding of evolutionary dynamics, bacterial infection, and the progression of inflammatory diseases.

A characteristic feature of bacterial colonization is the formation of large-scale order. Rod-shaped bacteria display nematic alignment on surfaces, wherein localized stress, surface friction, and elasticity trigger the formation of ordered domains and lead to the emergence of topological defects (*Doostmohammadi et al., 2016*; *Dell'Arciprete et al., 2018*; *Doostmohammadi et al., 2018*; *You et al., 2018*; *You et al., 2021*) and various types of self-assembled structures, including edge fingerings (*Farrell et al., 2013*) and vertical structures (*Beroz et al., 2018*; *Hartmann et al., 2019*).

In particular, ±½ topological defects are the typical orientational singularities observed among growing bacterial colonies and biofilms (*Doostmohammadi et al., 2016*; *Doostmohammadi et al., 2018*; *You et al., 2018*; *Yaman et al., 2019*). These topological defects have biological significance and regulate stress distribution across the structure, alter the physiology of the cells (*Saw et al., 2017*), and could control entire morphology; eventually, these effects trigger the formation of fruiting bodies (*Copenhagen et al., 2020*) and bacterial spores in biofilms (*Yaman et al., 2019*). Liquid crystal theory has successfully predicted the dynamics of these defects; $-\frac{1}{2}$ defects are stationary whereas $+\frac{1}{2}$ defects are generally motile (*Shankar and Marchetti, 2019*; *DeCamp et al., 2015*; *Giomi et al., 2013*). Another interesting structural order in bacterial colonies is anchoring, where the bacteria are tangentially oriented along the edge of the colony (*Su et al., 2012*; *Doostmohammadi et al., 2016*; *Dell'Arciprete et al., 2018*).

In this study, we assess the orientational dynamics of a crowded bacterial population competing for limited space. Unlike regular expanding colonies, if growing bacteria surround a closed area, domains of inward growth are formed. Under these conditions, entire mechanical interactions differ and lead to the formation of asters, formed as radially aligned +1 topological defects. With only a few exemptions (*Maroudas-Sacks et al., 2020*; *Meacock et al., 2021*), higher-order topological defects (*Thijssen and Doostmohammadi, 2020*; *Shankar et al., 2018*) are not commonly observed in extensile active matter systems, including growing bacterial colonies. These defects only appear under external modifications such as stress (*Rivas et al., 2020*), confinement (*Duclos et al., 2016*; *Opathalage et al., 2019*), and flow (*Martínez-Prat et al., 2019*). Our results also reveal that velocity profile is an important factor controlling the emergence of these radially aligned structures. Furthermore, we investigate the invasive advantages of this orientation for competing bacterial strains of different lengths.

Inward growth is commonly observed in various biological systems. During wound healing (*Basan et al., 2013*), cancer cell growth (*Lee et al., 2017*; *Vader et al., 2009*), and retina development (*Than-Trong and Bally-Cuif, 2015*; *Azizi et al., 2020*), similar dynamic mechanisms are underway. Our results may provide novel mechanistic insights into these dynamics, particularly on the physical conditions for radial structural alignments during these complex growth processes.

## Results
### Experimental observation of aster structures

To observe the dynamics of inward-growing bacterial colonies, we sparsely spread nonmotile *Escherichia coli* and *Bacillus subtilis* separately, on a flat agarose surface (see Materials and methods). Time-lapse fluorescence microscopy was then performed to investigate the temporal evolution of growing colonies. With colony growth, the closed area invaded by multiple colonies was observed across the plate. Rough colony interfaces gradually converge to symmetric, relatively smooth, and enclosed circular areas. We refer to these shrinking circular regions as inward-growing bacterial domains because the growth direction is toward the center of the area. *Figure 1* displays typical snapshots of the inward growth process (*Figure 1a and b*, *Video 1*, *Figure 1—video 1*). Unlike regular expanding colonies, the bacterial orientation around these domains is generally radial. To assess the orientation, we first analyzed the radial order parameter $S_R$ around the center of these domains. The radial order parameter $S_R$ can be expressed as:

$$\langle S_R \rangle = \frac{1}{N} \sum_i \cos\left[2\left(\theta^i - \phi^i\right)\right] \qquad (1)$$

where $\theta^i$ is the angular orientation with respect to x-axis and $\phi^i$ is the angular position of the bacterium $i$ in polar coordinates about the colony center. *Figure 1d and c* displays the bacterial orientation

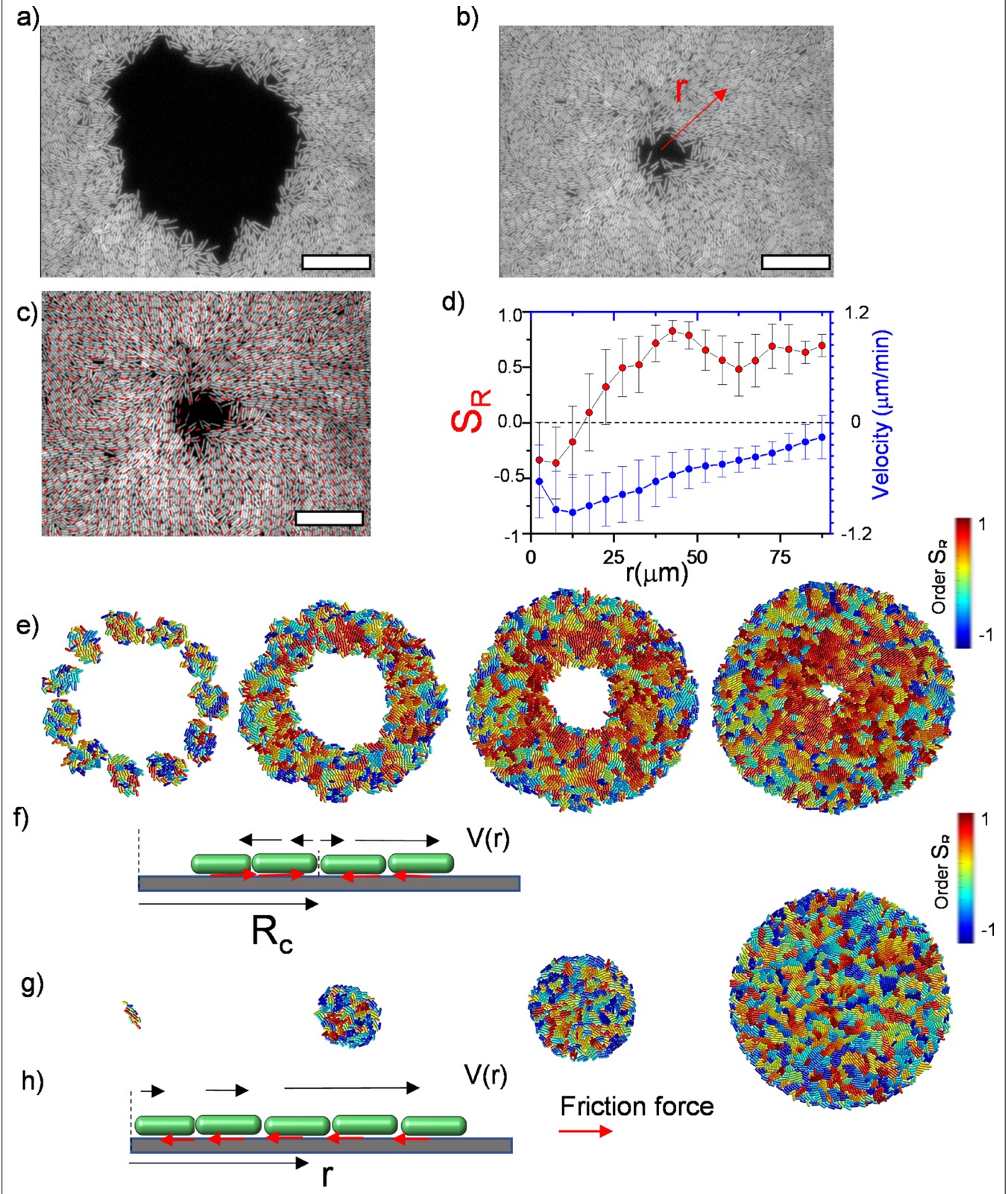

**Figure 1.** Experimental observation of inward growth of bacterial colonies and emergence of radial alignment. (**a**) Early stage of a closed area surrounded by growing bacterial colonies (*Bacillus subtilis*). (**b**) Snapshot of the radially aligned bacterial profile immediately before hole closure. (**c**) A director field superimposed on an inward-growing domain displaying radial alignment. Scale bar 25 μm. (**d**) The azimuthally averaged radial order parameter ($S_R$) and velocity against the distance from the colony center for the colony snapshot given in (**c**). Error bars are defined as s.d. (**e,g**)

*Figure 1 continued on next page*

*Figure 1 continued*

Simulation of 2D inward colony growth and regular expansion of bacterial colonies. (**f,h**) Schematic illustration of the velocity field (black arrow) and frictional force (red arrow) on bacteria in the inward and outward growing domain. Cell colors represent the radial order parameter ($S_R$). Red represents radial alignment; blue, tangential alignment.

The online version of this article includes the following video for figure 1:

**Figure 1—video 1.** Radial alignment during inward growth and emergence of asters.
https://elifesciences.org/articles/72187/figures#fig1video1

**Figure 1—video 2.** Two-dimensional (2D) simulation of the bacterial colony during regular expansion.
https://elifesciences.org/articles/72187/figures#fig1video2

and order parameter $S_R(r)$ as a function of radial distance. $S_R = +1$ corresponds to radial alignment and $S_R = -1$ corresponds to tangential alignment. It is evident that large-scale radial order emerges across these inward-growing domains (*Figure 1c and d*). These structures strongly resemble +1 topological defects also known as aster structures. We also measured the velocity of the bacterial flow during this process (*Figure 1d*). We found that the direction of the flow is toward the center. From these measurements, we can conclude this radial inward flow could align the bacteria in a radial direction.

## Numerical simulation of bacterial orientation during inward growth

To clarify the impact of flow-induced alignment and differences in orientation between inward-growing and regular expanding colonies, we simulated 2D bacterial growth using a hard rod model. We used the open-source simulation code GRO (*Jang et al., 2012*) which provides a fast platform to observe bacterial growth (see Materials and methods). To determine the morphology of the inward-growing domain, we initially distributed bacteria in a random orientation. With growth, bacteria form small colonies, which eventually fuse into a growing annulus (*Figure 1e*). To visualize the large-scale order, we color-coded bacteria on the basis of their radial orientation, with red representing radial orientation, and blue representing tangential orientation around the center of the hole (*Figure 1e*, *Video 2*). These simulation results captured the experimentally observed radial order across the colony.

However, regular expanding colonies only formed microdomains with random local orientations (*Figure 1g*, *Figure 1—video 2*). Regular expanding colonies represent the outward growth initiating from a single bacterium displayed in *Figure 1g*. Based on these simulations, the primary difference between regular expanding and inward-growing colonies is the sudden change in the direction of the surface drag force which depends on the velocity (*Figure 1f and h*). In inward-growing colonies, this force flips its sign at a critical radius where the local radial velocity of the colony vanishes.

To further quantify the effects of the critical radius, we determined the stress distribution and radial velocity profile $v_r$, in growing colonies. *Figure 2* summarizes the comparison and time

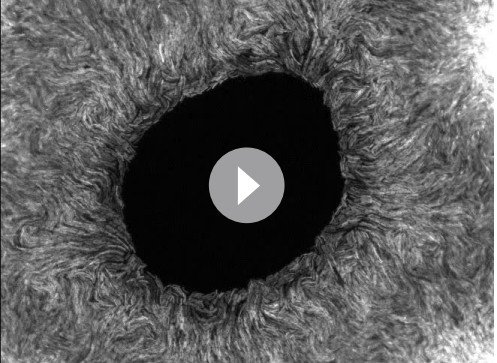

**Video 1.** Radial alignment during inward growth and emergence of asters. The video shows the fluorescence image of GFP labeled *Escherichia coli* (BAK 55) during inward growth. The duration of the experiment is 120 min and the total area is 125 × 125 μm².
https://elifesciences.org/articles/72187/figures#video1

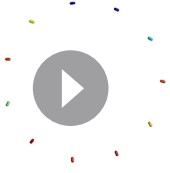

**Video 2.** Simulation of a bacterial colony during inward growth. This video is associated with Figure 1f.
https://elifesciences.org/articles/72187/figures#video2

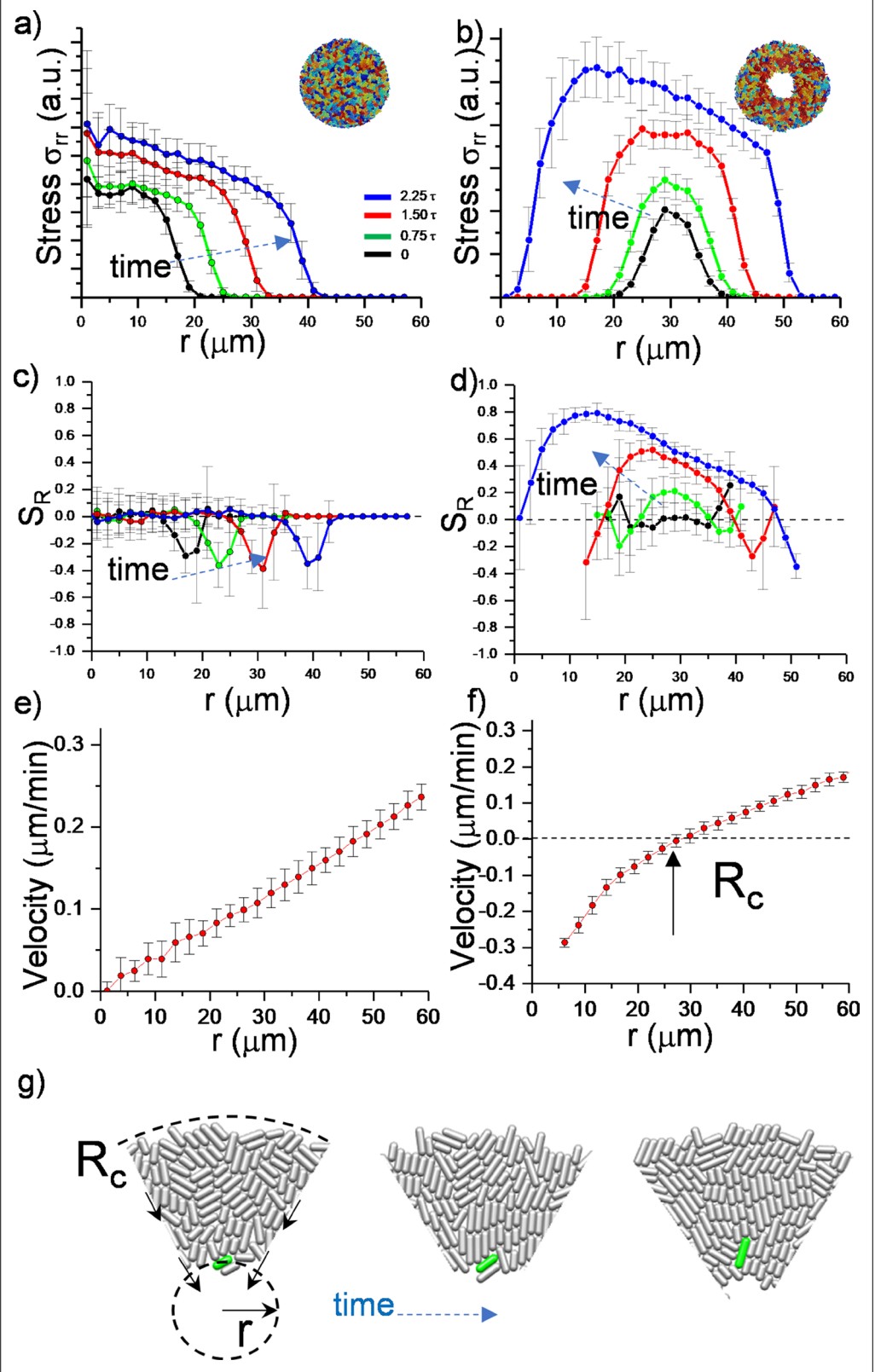

**Figure 2.** Numerical analysis of critical physical parameters during inward growth. (**a, b**) Plot of azimuthally averaged radial stress distribution ($|\sigma_{\mathrm{rr}}|$) at different time points of regular expanding and inward-growing colonies against the distances from the center of the colony. Time points are given in cell division time (*t*). (**c,d**) Plot of the azimuthally averaged radial order parameter ($S_R$) across the colonies. Radial order emerges not only below the

*Figure 2 continued on next page*

*Figure 2 continued*

critical radius but also beyond this level. Negative radial order corresponds to a tangential orientation or active anchoring. (**e,f**) Comparison of azimuthally averaged radial velocity ($v_r$) profiles of regular expanding and inward-growing colonies. Regular expanding colonies display a linear profile; however, inward-growing colonies form a nonlinear velocity profile. Error bars are defined as s.d. and averaged over 25 simulations. (**g**) Snapshots of gradual rotation of a single bacterium (green) into a radial orientation during inward growth. Similar bacterial rotation can be seen in experimental results (*Figure 2—figure supplement 5*).

The online version of this article includes the following figure supplement(s) for figure 2:

**Figure supplement 1.** Plot of azimuthally averaged stress distribution ($\left|\sigma_{\theta\theta}\right|$ and $\left|\sigma_{r\theta}\right|$) at different time points of regular expanding and inward-growing colonies against the distances from the center of the colony.

**Figure supplement 2.** Plot of azimuthally averaged packing fraction of bacteria at different time points of regular expanding and inward-growing colonies against the distances from the center of the colony.

**Figure supplement 3.** Molecular dynamics simulations of colony radius.

**Figure supplement 4.** Experimental measurements of the velocity profile.

**Figure supplement 5.** Experimental images of gradual rotation of a single bacterium (labeled green) into a radial orientation during inward growth.

**Figure supplement 6.** Simulation results of growing bacterial colony and scalar order parameters (*S*) under circular confinement defined by a fixed wall.

evolution of these parameters. We first focused on radial and azimuthal stress profiles. We noted that the stresses ($\left|\sigma_{rr}\right|$ and $\left|\sigma_{\theta\theta}\right|$) (*Figure 2—figure supplement 1*) are maximum around the critical radius during inward growth (*Figure 2a and b*). The stress profiles initially show the quadratic form which is particularly dictated by the radial velocity profile (*Beroz et al., 2018*).

Then we observed that, as colonies grew, only inward-growing colonies developed substantial radial order $S_R(r)$ (*Figure 2c and d*). Furthermore, radial velocity profiles $v_r$ significantly differed between regular expanding and inward-growing colonies. In contrast with regular expanding colonies, which have a linear radial velocity profile, inward-growing colonies developed radially nonlinear velocity, which vanishes at the critical radius (*Figure 2e and f*, *Figure 2—figure supplement 3*). Experimentally, similar profiles were measured (*Figure 2—figure supplement 4*). This profile gradually rotated the bacteria into the radial direction (*Figure 2g*, *Figure 2—figure supplement 5*). Based on these results, the velocity profile appears to be the key physical parameter regulating the flow-induced alignment and the formation of the radial order.

## Velocity profile and radial alignment

To better understand the association between the velocity profile on radial alignment, we first focused on the development of a minimum theoretical model based on active nematics. The theory of active nematics and liquid crystal physics provides a robust framework for understanding the dynamics of bacterial orientation. The primary characteristic of expanding colonies is the constant growth rate of the colony structure. The incompressibility criteria in 2D results in a linear relation between bacterial growth rate and radial velocity profile of the colony, $v_r = g(r) r = \frac{\Lambda}{2} r$, where $g(r)$ is the local growth rate and $\Lambda$ is the exponential bacterial growth rate. These coefficients are related to incompressible expanding bacterial colonies.

This relation was previously referred to as a Hubble-like constant owing to its similarity to the expansion of the universe (*Dell'Arciprete et al., 2018*). We considered the same approximations to obtain insights into bacterial orientation during inward growth. First, we used the assumption that without molecular field and convection terms, the time evolution of the orientational angle $\theta$ is simply regulated as follows (see Materials and methods):

$$\frac{d\theta}{dt} = \frac{\xi g' r}{2S} \sin\left(2\left(\phi - \theta\right)\right) \tag{2}$$

where $\phi$ is the angular position of the bacteria in polar coordinates, and $\xi$ is the flow alignment parameter. Furthermore, $g(r)$ is the local growth rate of the colony and its spatial derivative $g'(r)$ regulating the stability of the bacterial orientation. The constant growth rate observed in regular expanding colonies does not provide any orientational preference, $\frac{d\theta}{dt} = 0$. However, this condition significantly differs during inward growth, wherein the local growth rate can be expressed as follows:

$$g\left(r\right) = \frac{v_r}{r} = \frac{\Lambda}{2}\frac{\left(r^2 - R_c^2\right)}{r^2} \text{ and } g'\left(r\right) = \Lambda\frac{R_c^2}{r^3} > 0 \tag{3}$$

Our assumption of a constant critical radius (*Figure 2—figure supplement 3b*) indicates that the spatial derivative of the growth rate is positive everywhere across the colony, $g'\left(r\right) > 0$, suggesting the possibility of a stable state with $\theta = \phi$. The stable radial orientation stimulates aster formation, being referred to as a +1 topological defect. This finding is significant because $g'\left(r\right) \neq 0$ is generally possible in compressible structures and also only around leading edges of growing colonies due to sudden drop (*Dell'Arciprete et al., 2018*). Although bacterial colonies are not compressible, inward growth and the shrinking hole structure alter the overall velocity profile and lead to an essential local growth rate.

The radial orientation is stable throughout the colony and not only below the critical radius. To clarify this point, we simulated colony growth under a fixed circular wall mimicking the stationary critical radius (*Figure 2—figure supplement 6*). We observed a similar radial alignment. These results indicate the association between the circular confinement owing to the critical radius which dictates velocity profiles and the stability of bacterial orientations.

## Nemato-hydrodynamics and continuum modeling

Thereafter, we investigated whether the same defects were obtained through the continuum nemato-hydrodynamics equations of growing active matter (*Giomi et al., 2012*; *Olmsted and Goldbart, 1992*; *Mishra, 2017*). Due to coarse graining over specific physical details, the continuum model could provide generality of our observation. The model is based on continuity, Navier–Stokes equations, and dynamics of the order parameter tensor $\boldsymbol{Q}$ (see Materials and methods). The coupled differential equations governing the primary material fields density $\rho$, $\boldsymbol{Q}$, and velocity $\boldsymbol{\nu}$ can be expressed as follows:

$$\frac{D\rho}{Dt} = \Lambda\rho + D_\rho\nabla^2\rho \tag{4}$$

$$\frac{D\left(\rho\boldsymbol{\nu}\right)}{Dt} = \nabla\cdot\boldsymbol{\sigma} - \gamma\rho\boldsymbol{\nu} \tag{5}$$

$$\frac{D\boldsymbol{Q}}{Dt} = \xi\boldsymbol{u} + \boldsymbol{Q}\cdot\boldsymbol{\omega} - \boldsymbol{\omega}\cdot\boldsymbol{Q} + \Gamma^{-1}\boldsymbol{H} \tag{6}$$

Here, $\frac{D}{Dt}$ is the material derivative, and the stress tensor is given as:

$$\boldsymbol{\sigma} = -p\boldsymbol{I} - a\left(\rho\right)\boldsymbol{Q} - \xi\boldsymbol{H} + \boldsymbol{Q}\cdot\boldsymbol{H} - \boldsymbol{H}\cdot\boldsymbol{Q} \tag{7}$$

Here, $a\left(\rho\right)\boldsymbol{Q}$ represents the active stress originating from the extensile nature of bacterial growth. $\boldsymbol{u}$ and $\boldsymbol{\omega}$ are the traceless strain rate and vorticity, respectively (see Materials and methods). The critical parameter $\xi$ is the flow alignment parameter. The details of the frictional drag coefficient per unit density $\gamma$, the molecular field $\boldsymbol{H}$, pressure $p$, rotational diffusion constant $\Gamma$, and small diffusion coefficient $D_\rho$ are given in Material and methods. These equations were initially solved for growing bacterial colonies (*Volfson et al., 2008*; *Doostmohammadi et al., 2016*; *Dell'Arciprete et al., 2018*; *You et al., 2018*; *Atis et al., 2019*) and successfully predicted the active nematic nature and domain formations among colonies of rod-shaped bacteria. We solved them numerically with finite element method (FEM) (see Materials and methods). As a benchmark, we compared the simulations with regular expanding colonies. *Figure 3* summarizes the results of these continuum simulations. As expected, regular expanding colonies exhibited only local alignment (*Figure 3a and c*, *Figure 3—video 1*) corresponding to microdomains. However, inward-growing colonies developed robust radial alignment and order not only below but also beyond the critical radius (*Figure 3b and d*, *Video 3*). Inward-growing colonies also displayed the expected nonlinear velocity profile required for radial alignment (*Figure 3e and f*, *Figure 3—videos 2 and 3*, *Figure 3—figure supplement 1*). Moreover, a sudden drop of the velocity profile near the inner and outer colony edges also resulted in tangential orientation. These results from continuum simulation suggest that similar radial alignment could also be observed in other active matter systems under the same radial velocity profiles.

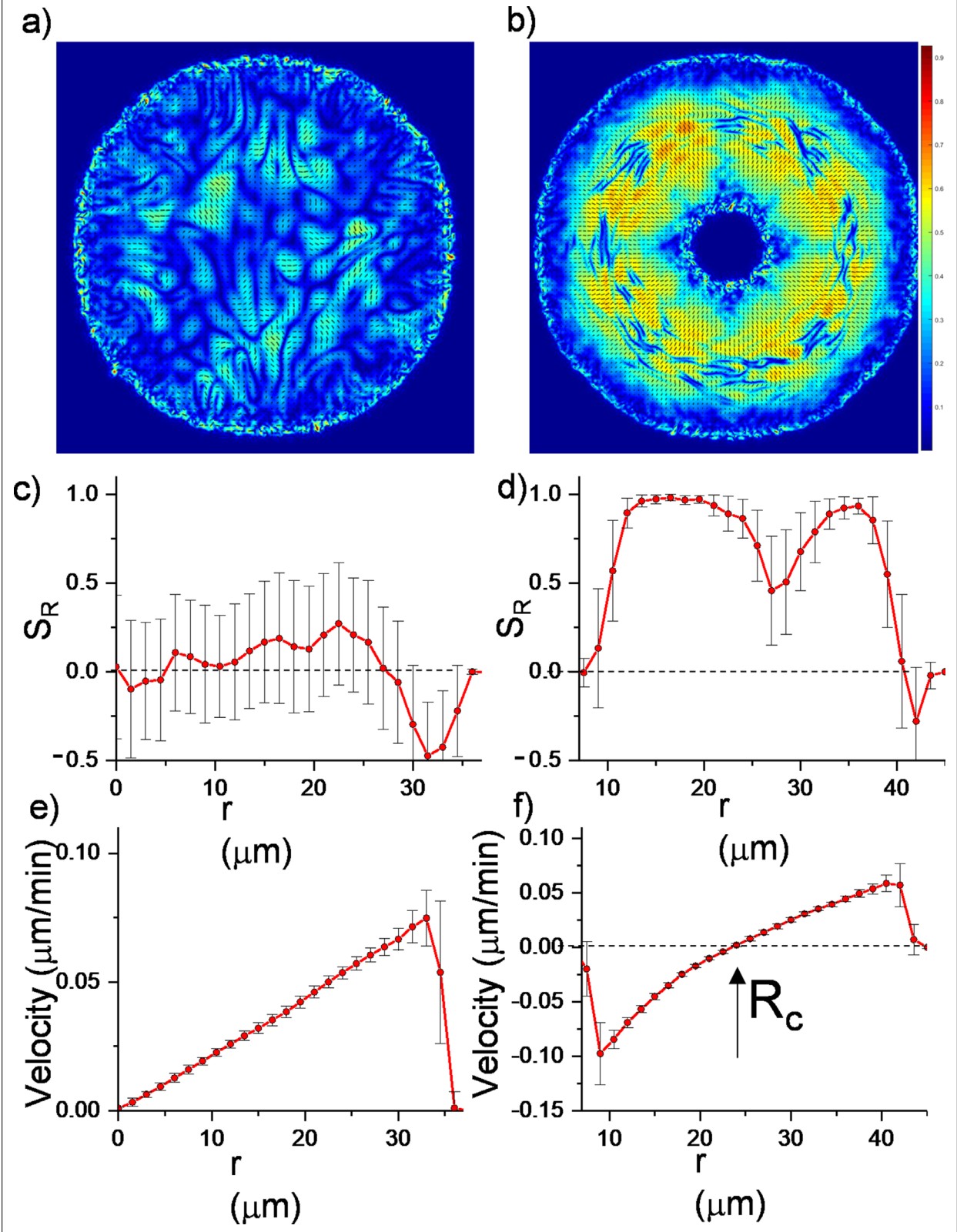

**Figure 3.** Continuum simulations of two-dimensional (2D) colony growth. (**a,b**) Scalar order parameter (*S*) overlapped with the director field pattern of both regular expanding and inward-growing bacterial colonies. Comparison of azimuthally averaged (**c,d**) radial order parameter (*S_R*) and (**e,f**) radial velocity (*v_r*) profiles of the colonies against the distance from the center of the colony. In contrast with regular expanding colonies, inward-growing

*Figure 3 continued on next page*

*Figure 3 continued*

domains developed radial order throughout the colony. A sudden velocity drop near the edge of the colonies resulted in tangential orientation. Error bars are defined as s.d.

The online version of this article includes the following video and figure supplement(s) for figure 3:

**Figure supplement 1.** Continuum simulation results of colony density and velocity profiles.

**Figure 3—video 1.** Continuum simulation of two-dimensional (2D) colony growth during regular expansion.
https://elifesciences.org/articles/72187/figures#fig3video1

**Figure 3—video 2.** Continuum simulation of two-dimensional (2D) colony growth during regular expansion.
https://elifesciences.org/articles/72187/figures#fig3video2

**Figure 3—video 3.** Continuum simulation of two-dimensional (2D) colony growth during inward growth.
https://elifesciences.org/articles/72187/figures#fig3video3

## Inward-growing domains in multi-layered colonies

Growing bacterial colonies on elastic substrates generally form multi-layered structures. We investigated whether these multi-layered structures could change the radial alignment during inward growth. Herein, experimentally we observed inward-growing domains only around the inner edge surrounded by dense multi-layered structures. This is because merging colonies and the accumulated stress trigger multi-layer formation and limit the size of monolayer region around the edge (*Figure 4—figure supplement 1*). We investigated whether these multi-layered structures affect the radial alignment during inward growth by performing three-dimensional (3D) FEM simulations based on recently developed algorithms (*Yaman et al., 2019*; *Vetter et al., 2015*; *Vetter et al., 2013*). Our previous computational tool (GRO) cannot simulate bacterial growth in 3D. Our FEM algorithms are relatively slow, but this approach is very powerful to capture detailed bacterial growth in three-dimentional complex environments. The bacterial cells were modeled as growing elastic rods that undergo controlled cell division during colony growth. We first tested the 3D capability of FEM simulations by replicating similar radial alignment under spherical confinement (*Figure 4—figure supplement 2*). Then we focused on growing colonies on flat surfaces with surface friction. *Figure 4* shows the prototypical FEM simulation outcome from inward-growing colonies. As expected, accumulated stress triggers verticalization and multi-layer formation around the critical radius of the colony (*Figure 4a and b* and *Video 4*). However, a bacterial monolayer was observed only around the inner and outer leading edges of the colony. The formation of a monolayer region around growing colonies has been investigated in great detail (*Warren et al., 2019*). We found that these monolayers could also result in planar radial alignment (*Video 5*). The width of the monolayer was approximately $\Delta r$ = 90±30 µm (*Figure 4c*). This width defines the size of the aster structures observed herein.

## Inward-growing domains in monolayer colonies

So far, we experimentally studied naturally emerged inward-growing domains on agar surfaces. These domains are randomly formed across the plate. Due to random seeding of bacteria, outward growing edges of multiple colonies merge and form multi-layered structures. In these experiments particularly, the confinement is defined by crowded multi-layered environments. Thus, observing critical radius, outer growing edge and detailed velocity profiles are not possible around these dense regions. Our simulations showed that the initial annulus shape could overcome these limitations. We asked whether we could induce similar annulus structures by patterning the initial distribution of bacteria to observe both inward and outward

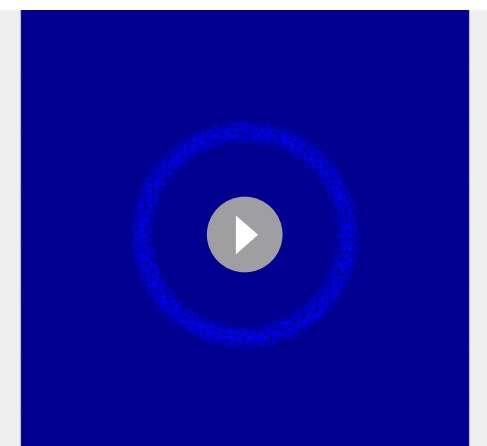

**Video 3.** Continuum simulation of two-dimensional (2D) colony growth during inward growth. The scalar order parameter is overlapped with the director field pattern. This video is associated with Figure 3b.
https://elifesciences.org/articles/72187/figures#video3

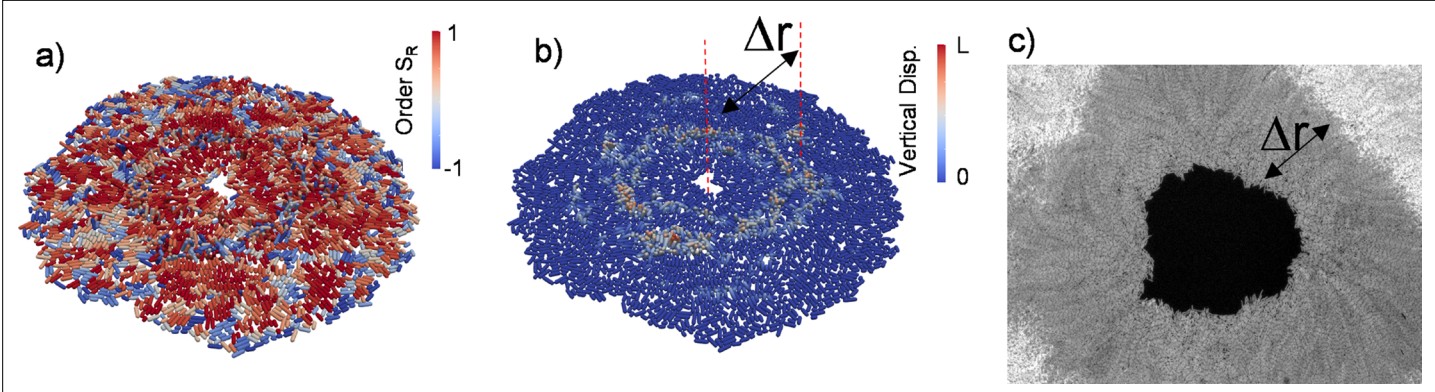

**Figure 4.** Three-dimensional (3D) colony growth and multi-layer formation. (**a,b**) Snapshot of inward-growing bacterial domains obtained through the finite element model simulation in 3D. The colors represent (**a**) the radial order parameter ($S_R$) and (**b**) the vertical displacement of the bacteria during growth. *L* is the bacterial length. (**c**) Experimental snapshot of the inward-growing domain indicating the transition between mono- to multi-layer. Δr represents the width of the monolayer bacteria domain.

The online version of this article includes the following figure supplement(s) for figure 4:

**Figure supplement 1.** Comparison of the formation of multi-layered colony structures with and without physical confinement.

**Figure supplement 2.** Three-dimensional (3D) finite element method (FEM) simulation results of growing bacterial colony and radial order for (**a**) freely expanding bacterial colony and (**b**) inward-growing colony under spherical confinement.

growing domains. We first tried to imprint bacteria on an agarose surface using soft PDMS molds. However, the wet surface and capillary effect quickly disturbed the initial bacterial patterns defined by the mold. Then we preferred non-contact lithographic techniques for patterning. Using a photomask (*Figure 5—figure supplement 1*), we exposed randomly distributed bacteria with blue light to define an initial growth geometry by killing the remaining part of the pattern (see Materials and methods). *Figure 5a* shows the time evolution of growing bacteria starting from annulus-shaped initial distribution. We observed that on a regular agar surface, again multi-layer formation dominates the overall colony morphology. Only very narrow monolayer regions are observable around the inner and outer edges of the colony. We then focus our attention on how to eliminate this multi-layering process. A simple glass or PDMS confinement cannot eliminate this multi-layering (*Figure 4—figure supplement 1*). Previous studies showed that attractive biochemical interactions between bacteria and surface could generate additional strong friction force (*Duvernoy et al., 2018*). Altogether friction force, stress accumulation, and verticalization of bacteria in a monolayer colony trigger the formation of these multi-layered structures. This process is mainly controlled by the competition between vertical

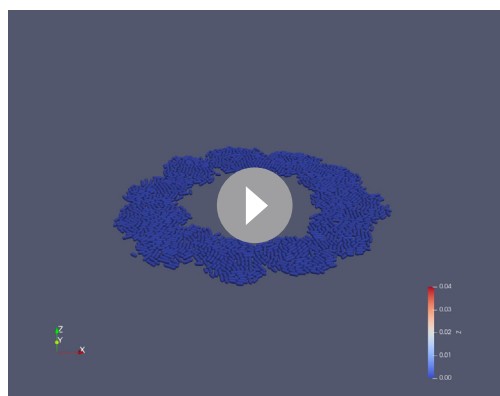

**Video 4.** Finite element method (FEM) simulation of colony growth during inward growth. Color represents vertical displacement. This video is associated with Figure 4b.

https://elifesciences.org/articles/72187/figures#video4

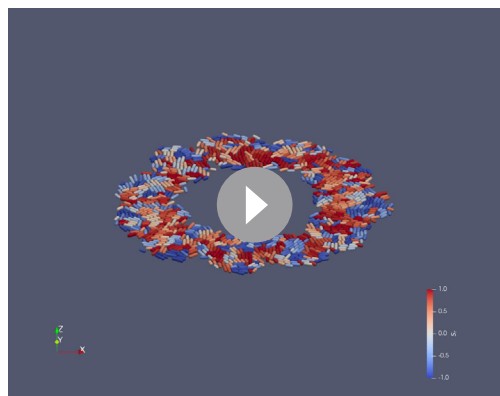

**Video 5.** Finite element method (FEM) simulation of colony growth during inward growth. Color represents the radial order parameter. This video is associated with Figure 4a.

https://elifesciences.org/articles/72187/figures#video5

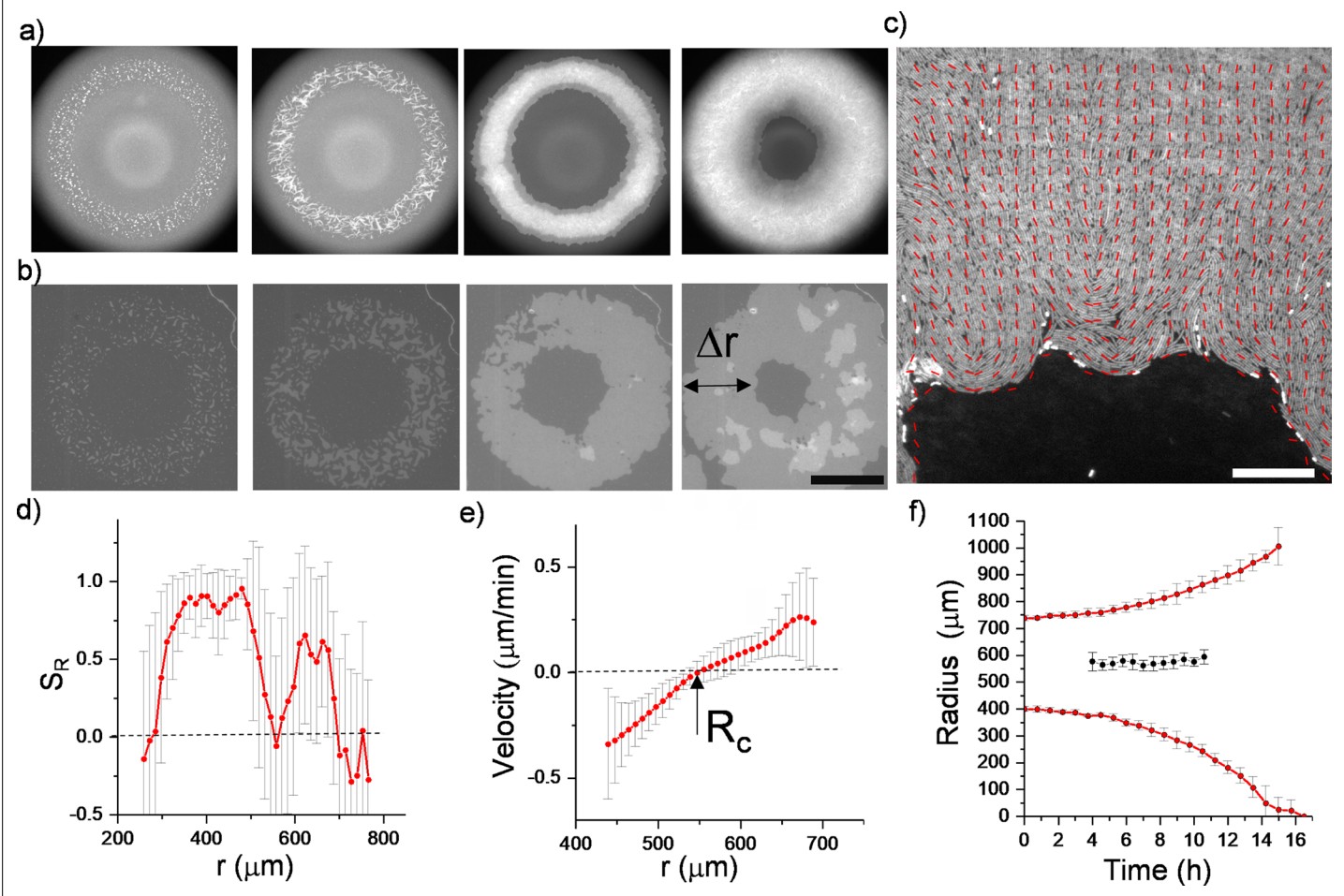

**Figure 5.** Patterning initial distribution of growing bacterial colonies. (**a,b**) Snapshot of inward-growing bacterial colonies (*Bacillus subtilis*) on different surfaces. Initial bacterial distribution is defined by non-contact lithographic techniques with blue light exposure. Inward-growing domains are formed by merging colonies originating from annulus-shaped initial distribution. (**a**) On an agarose surface, growing colonies easily form multi-layer structures and provide narrow monolayer inward-growing domains. However, (**b**) on a low friction polycarbonate (PC) surface, bacteria form monolayer colonies and provide large-scale inward-growing domains. Scale bar 500 µm. (**c**) Magnified fluorescence image superimposed with director field of the inward-growing domain indicating the radial alignment of bacteria. $\Delta r$ represents the width of the monolayer region. Scale bar 50 µm. (**d**) Radial order parameter ($S_R$) and (**e**) radial velocity profile ($v_r$) as a function of distance across the monolayer colony. Velocity profile was extracted by using PIV algorithms around the center of the annulus shape (*Figure 5—video 2*). (**f**) The experimentally measured inner and outer and critical radius. d, e, f are averaged over four different colonies, starting from the same initial annulus-shaped distribution.

The online version of this article includes the following video and figure supplement(s) for figure 5:

**Figure supplement 1.** Photomask with different annulus shapes was used to define the initial distribution of bacteria.

**Figure supplement 2.** Snapshot of inward-growing bacterial domains with the color representing compressive stress (**a**) on individual bacteria, velocity profile (**b**), and azimuthally averaged $|\sigma_{rr}|$ radial, $|\sigma_{\theta\theta}|$ hoop stress, the velocity profile, and the positions of the first five extrusion events triggering the multi-layering process (**c**).

**Figure supplement 3.** Sample SEM images of growing colonies on a polycarbonate (PC) membrane surface.

**Figure supplement 4.** Fluorescence image of growing monolayer colony on a polycarbonate (PC) surface during inward growth.

**Figure supplement 5.** Sample fluorescence image of inward-growing domain superimposed with director field showing large-scale radial bacterial alignment.

**Figure supplement 6.** Fluorescence image of growing monolayer colony on a polycarbonate (PC) surface starting from a single bacterium.

**Figure supplement 7.** Sample PIV measurement used to determine the position of stationary regions indicating the critical radius $R_c$.

**Figure supplement 8.** Radial alignment of biofilm-forming bacteria during inward growth.

**Figure 5—video 1.** Time evolution of growing colonies starting from annulus-shaped initial distribution.
https://elifesciences.org/articles/72187/figures#fig5video1

*Figure 5 continued on next page*

*Figure 5 continued*

**Figure 5—video 2.** High magnification time-lapse imaging of bacterial growth around the critical radius.
https://elifesciences.org/articles/72187/figures#fig5video2

**Figure 5—video 3.** Time evolution of growing biofilm-forming colonies starting from annulus-shaped initial distribution.
https://elifesciences.org/articles/72187/figures#fig5video3

**Figure 5—video 4.** Two-dimensional (2D) finite element method (FEM) simulation results of growing biofilm structure during inward growth.
https://elifesciences.org/articles/72187/figures#fig5video4

---

force and lateral compression in the colony (*Grant et al., 2014*; *Beroz et al., 2018*; *Duvernoy et al., 2018*; *You et al., 2019*). Above the critical stress level, the orientation of rod-shaped bacteria becomes unstable and triggers the extrusion. Performing FEM simulations, we noticed that this extrusion process occurs around the center of the annuls and it can be controlled by surface friction (*Figure 5—figure supplement 2*). Although we don't know the detailed biological mechanism behind the friction force, it is evident that minimizing the surface friction can increase the size of the monolayer colony. Then, we tested the same bacterial patterning on different membranes to find a surface with low friction by minimizing biochemical interaction. We noted that only polycarbonate (PC) surfaces are useful for this purpose, and they support large monolayer colonies while providing a sufficient bacterial growth rate (*Figure 5—figure supplement 3*, *Figure 5—video 1*, see Materials and methods). The size of these monolayer colonies was approximately 600 µm. As we observed in our previous simulations (*Figure 4*), at a later stage, the second layer formation appeared around the center of these annulus shapes which is close to the critical radius (*Figure 5a*, *Figure 5—video 2*). Similarly, we observed strong radial alignment across the colony (*Figure 5c and d*, *Figure 5—figure supplements 4 and 5*). We did not observe any radial alignment in regular isolated monolayer colonies. Instead, we clearly observed orientational defects and microdomains in these monolayer colonies on PC surfaces (*Figure 5—figure supplement 6*). Inward and outward growing monolayer domains also provided the nonlinear velocity profile (*Figure 5e*), which is essential for radial alignment. We noticed that during this process critical radius shows a constant profile (*Figure 5f*, *Figure 5—figure supplement 7*, *Figure 5—video 2*). The other interesting form of bacterial growth is biofilm formation which has filamentous and nematic internal structures. As a next step, we similarly tested the radial alignment dynamics of these bacterial biofilms during the inward growth process, starting from the same initial distribution. We used a biofilm-forming strain *B. subtilis* 168 (*Yaman et al., 2019*) and observed similar strong radial alignment across the biofilms (*Figure 5—video 3*, *Figure 5—figure supplement 8*). Our FEM simulations also captured the alignment process of growing elastic bacterial biofilm structures (*Figure 5—video 4*, *Figure 5—figure supplement 8c,d*).

## Biological significance

Finally, to assess the biological significance of radial alignment, we investigated whether these structures potentially affect the competition among bacteria during inward growth. In general, near the leading edge of a bacterial colony, competition strongly depends on physical parameters. The most prominent example is a genetic drift based on random fluctuations (*Hallatschek et al., 2007*; *Kayser et al., 2018*). This phenomenon could be altered through steric interactions among the cells, which can potentially alter the evolutionary dynamics of competing bacteria (*Farrell et al., 2017*). Although bacterial orientation is generally tangential at the expanding colony edge, radial bacterial alignment potentially contributes to inward growth. We hypothesized that longer rod-shaped bacteria potentially have an advantage owing to the torque balance. The basic premise is that the torque depends on the length of the bacteria, resulting in rapid radial alignment. Radial alignment further leads to lane formation and promotes an invasive advantage to the longest one, which could be beneficial in terms of approaching nutritional hotspots localized around the defect core more effectively.

To assess this competition, we initially simulated the growth dynamics of a mixed population with different division lengths from the same random initial distribution on a circle (*Figure 6—figure supplement 1*). This is the most challenging condition to test the impact of the length difference on the bacterial alignment. Although the initial distribution of the bacteria is random, long bacteria can develop a higher radial order during inward growth (*Figure 6a–f*). This radial order gradually allows the longest bacteria to approach the center of the defect more effectively (*Figure 6g*). The bacterial

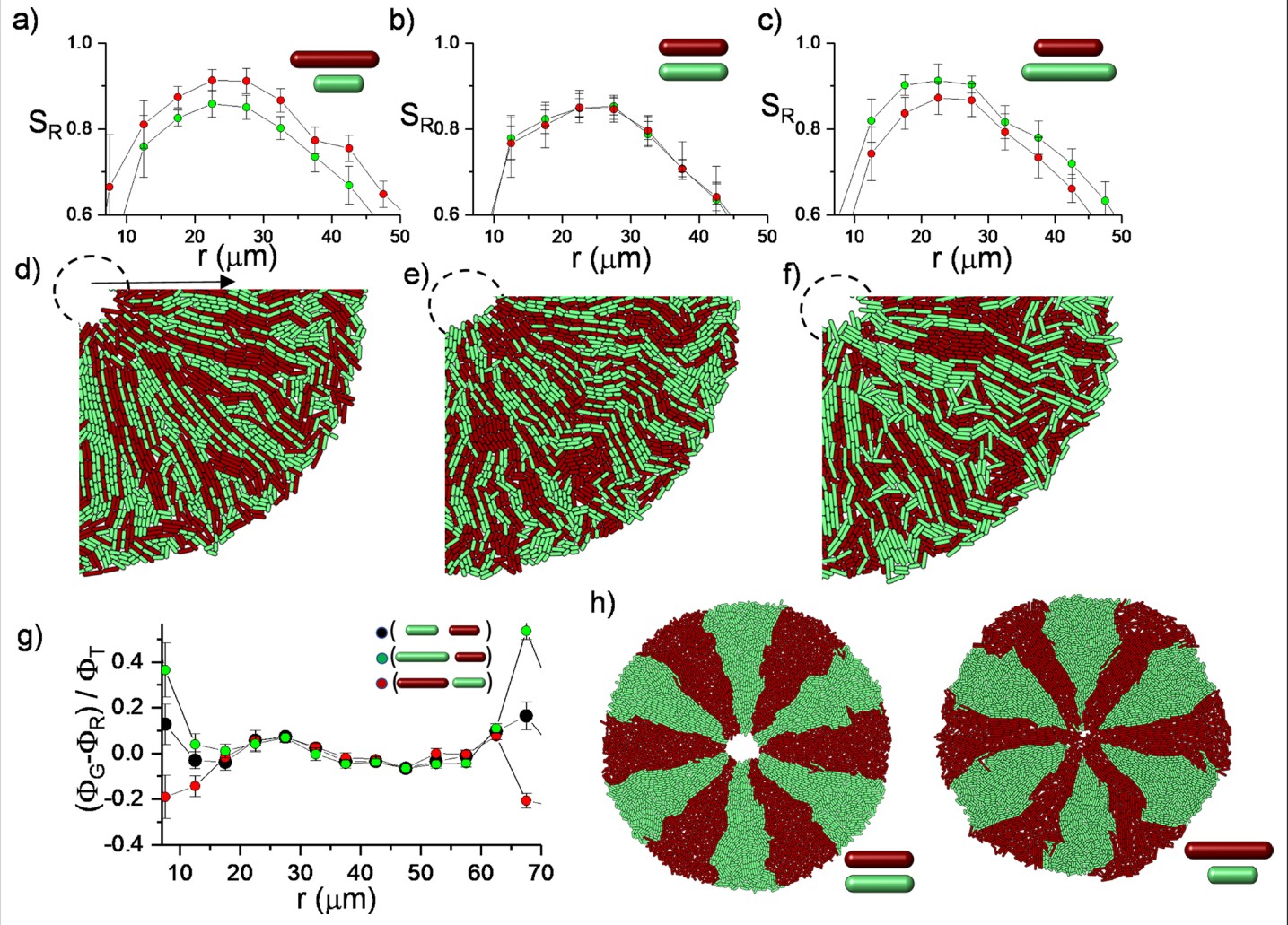

**Figure 6.** Simulations of bacterial competition during inward growth.

The online version of this article includes the following figure supplement(s) for figure 6:

**Figure supplement 1.** Two-dimensional (2D) simulation results of competing bacterial strains with different division lengths.

**Figure supplement 2.** Experimental verification of bacterial segregation due to random fluctuation during inward growth.

**Figure supplement 3.** Simulation results of competition of segregated bacterial strains.

growth is local, and it can create strong segregation within the colonies. The impact of the length could be more significant in segregated colonies. To visualize the difference, we initially segregated the bacterial strains with different lengths around the edge of the colony. Similar segregation can be commonly observed around the edge of the colony owing to random fluctuations. These segregations can also occur during inward growth (*Figure 6—figure supplement 2*). Instead of expanding segments owing to perimeter inflation, we observed shrinking segments owing to the deflation of the hole geometry. Computationally, the advantage of radial alignment was more evident in segregated bacterial colonies (*Figure 6h*). Interestingly, in a monolayer colony, radial alignment promotes the invasion of both the center and the leading outer edge of the colony through the longest bacteria (*Figure 6g*). After the complete invasion of the center, the radial lanes buckle (*Figure 6—figure supplement 3*, *Videos 6 and 7*). However, experimental verification of this competition remains challenging. Although precise regulation of the aspect ratio of bacterial morphology is well established (*Dion et al., 2019*), cell length can still not be independently tuned without perturbing other essential physiological properties, including growth rate and the biofilm-forming potential of the bacteria.

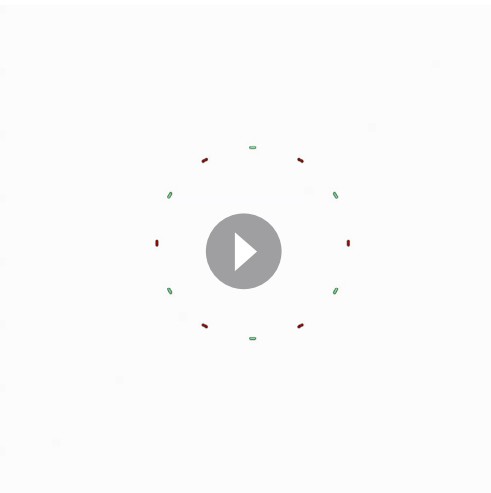

**Video 6.** Simulation results of competing bacterial strains with identical division lengths. RFP and GFP labeled bacteria are initially segregated. Color represents the bacterial type. This video is associated with Figure 6h.

https://elifesciences.org/articles/72187/figures#video6

## Discussion

Radially aligned structures can be considered as a +1 aster defect. These are ubiquitous topological structures observed in biological (*Roostalu et al., 2018*; *Ross et al., 2019*; *Kruse et al., 2004*; *Julicher et al., 2007*) or synthetic (*Sokolov et al., 2019*; *Snezhko and Aranson, 2011*) active matter systems. For instance, microtubules can form nematic alignment or asters during mitosis, depending on the extensile or contractile activity. Bacterial colonies can be considered an extensile active material platform, generally supporting the formation of only ±½ topological defects. This study shows that stable radially aligned, aster structures can also emerge during inward growth. In particular, we report the critical role of the colony velocity profile during this process, which depends on numerous factors. Although the bacterial growth rate is constant throughout the colony, growth geometry, confinement, or boundary conditions can alter the velocity profile. Together, these biomechanical interactions change the bacterial orientation and stability, thus generating ordered structures. Different types of ordered structures have been observed in bacterial biofilms (*Yan et al., 2016*) and 3D colonies (*Warren et al., 2019*). Furthermore, we believe that the velocity profile of growing structures on flat surfaces plays a significant role in bacterial alignment. Future studies are required to investigate the contribution of these effects.

We should emphasize that inward-growing bacterial colonies and wrinkling thin circular sheets have geometric similarities (*Davidovitch et al., 2011*). In these elastic circular objects, under axisymmetric tensile load, azimuthal stress (hoop stress, $\sigma_{\theta\theta}$ ) show transition from tensile to compressive profile which eventually creates radial wrinkling pattern below critical radius. However, unlike elastic objects, growing bacterial colonies can only develop compressive stress due to negligible attractive force between bacteria. Experimental measurement of internal stress could provide more details, but it remains challenging. We noticed that the packing fraction of the bacteria shows a correlated profile (*Figure 2—figure supplement 2*). However, particularly for aligned bacteria, it is still very difficult to extract this information. In the future, new molecular probes could be useful for the experimental measurement of accumulated stress in the bacterial colonies (*Chowdhury et al., 2016*; *Prabhune et al., 2017*).

Finally, this study reveals the potential biological significance of radial alignment during the invasion. These ordered structures provide additional advantages and promote the survival of the longest bacteria. These results link the orientational properties and competition dynamics of bacterial colonies. Our findings are of potential relevance for the understanding of complex dynamics of bacterial infections and the progression of inflammatory diseases.

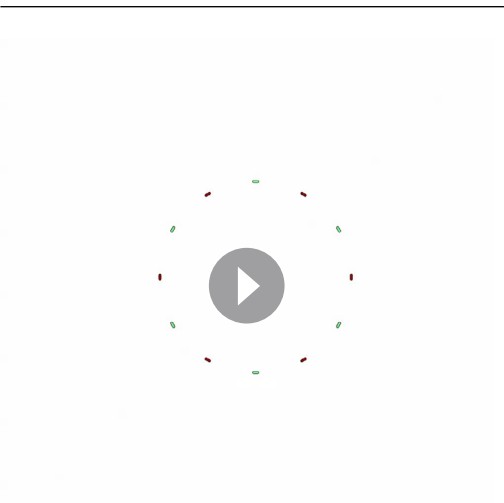

**Video 7.** Simulation results of competing bacterial strains with different division lengths. RFP and GFP labeled bacteria are initially segregated. Color represents the bacterial type. This video is associated with Figure 6h.

https://elifesciences.org/articles/72187/figures#video7

**Table 1.** List of strains used in this study.

| Strain | Parent | Operation | Genotype |
|---|---|---|---|
| BAK47 | 168 | Transformed with plasmid ECE321 from Bacillus Genetic Stock Center | *amyE*::Pveg-sfGFP (Spc) |
| BAK115 | TMN1138 | Transformed with plasmid ECE327 from Bacillus Genetic Stock Center | *amyE*::Pveg-mKate (Spc) *sacA*::P$_{hag}$-mKate2L (Kan) *hagA233V* (Phleo) |
| BAK51 | TMN1138 | Transformed with plasmid ECE321 from Bacillus Genetic Stock Center | *amyE*::Pveg-sfGFP (Spc) *sacA*::P$_{hag}$-mKate2L (Kan) *hagA233V* (Phleo) |
| BAK 55 | DH5alpha | Transformed with plasmid 107741 from Addgene | *pDawn-sfGFP* |

# Materials and methods

## Bacterial preparation and growth conditions

Bacterial cultures (BAK47 and BAK51) were grown in Luria-Bertani (LB) broth at 37°C on a shaker. An overnight culture was diluted 100× and grown for 8 hr. The culture was diluted 10,000×, and 10 µl of culture was seeded on an LB agarose plate. These isolated bacteria on plates were grown at 21°C for 12 hr and then imaged. Strains used in experiments are described in *Table 1*. In *B. subtilis* bacterial strains, the flagella-producing gene (*hag*) was mutated to eliminate the swimming-induced motion. The background strain TMN1138 was obtained from R Losick Lab.

## Microscopy imaging

Fluorescence time-lapse imaging was performed using a Nikon inverted and Stereo SMZ18 microscopes. Images were obtained using a Andor EMCCD camera. Time intervals between successive images were set to 5–10 min.

## 2D hard-Rod simulations of a growing colony

We used the open-source simulation program GRO based on a hard-rod model. The code is available from https://depts.washington.edu/soslab/gro/. We modified the original code to be able to change the initial bacterial position and to extract the orientation of the bacteria. Sample files and scripts used for modifications are available on GitHub (https://github.com/mustafa-basaran/Large_Scale_Orientation_Bacteria, copy archived at swh:1:rev:fd7673254fa57874676b183f9a944d9e457c3ac0; *Basaran, 2021*).

## Bacterial patterning

We used photolithographic techniques to define the initial distribution of the bacteria by killing with structured blue light illumination. We used 15 min exposure under 5 mW/mm² 480 nm uniform light beam. We think the killing mechanism is mainly based on the local drying process. The geometry was defined by chromium photomask. The mask (*Figure 5—figure supplement 1*) was fabricated by using Heidelberg DWL 66+ Direct Writing Lithography System and developed with chromium etchant. We have tested different annulus-shaped patterns by tuning the inner and outer radius. Due to light diffraction, the final exposed pattern depends on the spacing between the PC filter and a mask. Our optimized pattern has 400 µm inner and 800 µm outer radius.

## Growing monolayer colonies on low friction surfaces

In order to minimize surface friction, we replaced the agarose surface with a filter membrane. We have tested several membrane filters, including nylon, polycarbonate (PC), polyethersulfone, cellulose acetate. Only white PC filters with 0.4 µm pore size supported stable and large-scale monolayer colony formation. We noted that the brown PC filter has similar low surface friction; however, it has strong light absorption and does not allow noncontact lithography due to heavy condensation on the photomask.

## SEM imaging

A PC filter paper with a pore size of 0.4 µm was placed on LB agar surface. After seeding the 10,000× diluted bacteria on the filter paper, bacteria were grown on the paper for 12 hr at 21°C. Then the filter

paper was peeled off from the surface, and the colonies were fixed using paraformaldehyde and left to dry. Fixed colonies were coated with 20 nm gold and imaged using a Zeiss Ultra Plus Field Emission Electron Microscope.

## Calculating stress distribution in a growing colony

The stress inside the colony can be calculated from the virial expansion (**Volfson et al., 2008**) $\sigma_i = \frac{1}{2a_i'}\sum_j r_{ij}F_{ij}$ , $a_i'$ is the effective area, $r_{ij}$ is the position of the contact, and $F_{ij}$ is the interaction force between the cells. Using the following transformations, we calculated the stress in polar coordinate

$$\sigma_{rr} = \sigma_{xx} * \cos^2(\theta) + \sigma_{yy} * \sin^2(\theta) + \sigma_{xy} * \sin(\theta) * \cos(\theta)$$

$$\sigma_{\theta\theta} = \sigma_{xx} * \sin^2(\theta) + \sigma_{yy} * \cos^2(\theta) + \sigma_{xy} * \sin(\theta) * \cos(\theta)$$

$$\sigma_{\theta r} = \cos(\theta)\sin(\theta)(\sigma_{yy} - \sigma_{xx}) + \sigma_{xy} * \cos(2 * \theta)$$

Due to negligible lateral friction between bacteria, we ignore $\sigma_{xy}$ and our equations become:

$$\sigma_{rr} = \sigma_{xx} * \cos^2(\theta) + \sigma_{yy} * \sin^2(\theta)$$

$$\sigma_{\theta\theta} = \sigma_{xx} * \sin^2(\theta) + \sigma_{yy} * \cos^2(\theta)$$

$$\sigma_{\theta r} = \cos(\theta)\sin(\theta)(\sigma_{yy} - \sigma_{xx})$$

## 3D FEM simulations of growing bacterial colonies

For the 3D computer simulations, we employed an open-source (https://libmesh.github.io/) parallel finite element library written in C++ (**Vetter et al., 2013**). Sample files and scripts used for analysis are available on GitHub (https://github.com/mustafa-basaran/Large_Scale_Orientation_Bacteria, copy archived at swh:1:rev:fd7673254fa57874676b183f9a944d9e457c3ac0; **Basaran, 2021**). Analogous to **Yaman et al., 2019**, the bacteria were modeled as an isotropic, linearly elastic continuum whose initial stress-free shapes were spherocylinders. The bacteria were assumed to maintain a uniform circular cross-section with radius $r = 0.5$ m, a mass density of $1\,\mathrm{gcm}^{-3}$ , a Young's modulus of $E = 5300\mathrm{Pa}$, and a Poisson ratio of $\nu = 1/3$. The total elastic energy $U$ of each bacterium comprised the usual terms for axial dilatation or compression, bending, and torsion (**Vetter et al., 2015**):

$$U = \frac{E\pi r^2}{2}\int_0^L \varepsilon^2 + \frac{r^2}{4}\left(\kappa^2 + \frac{1}{1+\nu}\varphi^2\right)ds$$

where $L$ denotes the bacterium length, $\varepsilon$ the axial Cauchy strain, $\kappa$ the scalar midline curvature, and $\varphi$ the twist per unit length. Hertzian steric forces were exchanged between overlapping bacterial elements in a normal direction. Tangential forces and torques exchanged during contact between bacterium pairs and between bacteria and the substrate was computed with a slip-stick friction model with a uniform isotropic Coulomb friction coefficient. We modeled the substrate as an elastic half-space onto which the bacterial colony was placed, and exerted a perpendicular gravitational force on the bacteria. The bacteria were grown exponentially in length over time by continuously increasing each element's equilibrium length. For this study, the finite element program was extended to allow for cell division when the bacterial length surpassed a division threshold $L_\theta = 5$ m. When $L > L_\theta$ , the bacteria were split into two pieces at a random position drawn from a normal distribution about their center with a standard deviation of $L/10$, but no further away from the center than $L/5$. To evolve the colony in time, Newton's translational and rotational equations of motion were integrated with a Newmark predictor-corrector method of second order. To equilibrate the colony during growth, viscous damping forces were added.

In order to simulate inward-growing biofilm structure, we used our previous biofilm model (**Yaman et al., 2019**) and the same parameters. Eight identical replica of small biofilm structures are circularly distributed to form an initial annulus shape. We used fracture strain (0.3) to relax the extreme bending condition by triggering filament division.

## Radial velocity profile

To calculate radial velocity profile, $v_r$, during inward growth, we assume there is a critical radius $R_c$, where $v_r$ is equal to zero. For $r < R_c$ and the initial domain size equals to $A_{in}$:

$$A(t) = A_{in}e^{\Lambda t} = \pi \left(R_c^2 - r^2\right) \tag{8}$$

If we take derivative wrt time.

$$A_{in}\Lambda e^{\Lambda t} = -2\pi r \frac{dr}{dt} \tag{9}$$

$$v_r(r,t) = \frac{dr}{dt} = -\frac{\Lambda A(t)}{2\pi r} = \frac{\Lambda \left(r^2 - R_c^2\right)}{2r} \tag{10}$$

For outer growth where $r > R_c$

$$A(t) = A_{out}e^{\Lambda t} = \pi \left(r^2 - R_c^2\right) \tag{11}$$

Similarly, the time derivative is

$$A_{out}\Lambda e^{\Lambda t} = 2\pi r \frac{dr}{dt} \tag{12}$$

$$v_r(r,t) = \frac{dr}{dt} = \frac{\Lambda A(t)}{2\pi r} = \frac{\Lambda \left(r^2 - R_c^2\right)}{2r} \tag{13}$$

which results in the same equation. Consider that for $r$ lower than $R_c$, velocity will be negative (inward direction) and for $r$ greater than $R_c$ velocity will be positive (outward direction).

## Continuum modeling

For the continuum modeling, *Equations 4–7* were solved with the FEM in COMSOL Multiphysics. The material derivative is given by $D/Dt = \partial_t + v \cdot \nabla + (\nabla \cdot v)$. $u$ and $\omega$ are the strain rate and vorticity tensors, respectively, with components $u_{ij} = (\partial_i v_j + \partial_j v_i - \delta_{ij}\nabla \cdot v)/2$ and $\omega_{ij} = (\partial_i v_j - \partial_j v_i)/2$. We constructed a traceless and symmetric $Q$-tensor field:

$$Q_{\alpha\beta} = S\left(n_\alpha n_\beta - \frac{1}{2}\delta_{\alpha\beta}\right) \tag{14}$$

and we defined scalar order parameter:

$$S(r) = 2\sqrt{Q_{xx}^2(r) + Q_{xy}^2(r)} \tag{15}$$

The molecular field $H$ can be obtained starting from the Landau-de Gennes free energy density given as:

$$f_{LdG} = \frac{1}{2}K\left|\nabla Q\right|^2 + \frac{1}{2}\alpha(\rho)\,\text{tr}\left[Q^2\right] + \frac{1}{4}\beta(\rho)\left(\text{tr}\left[Q^2\right]\right)^2 \tag{16}$$

Therefore, $H = \delta/\delta Q \int dA f_{LdG} = K\nabla^2 Q - \alpha(\rho)Q - \beta(\rho)\,\text{tr}\left[Q^2\right]Q$. In the simulations, the relationships $a(\rho) = a_0\rho$, $\beta(\rho) = \frac{\alpha_0}{2}\rho$, $\alpha(\rho) = \alpha_0(\rho_c - \rho)$, and $p = G * max\left\{\left(\frac{\rho}{\rho_0} - 1\right), 0\right\}$ were used. We set the initial cell density $\rho_0 = 1$, growth rate $\Lambda = 0.005$, frictional drag coefficient per unit density $\gamma = 0.2$, flow aligning parameter $\xi = 0.7$, rotational diffusion constant $\Gamma = 1$, and the remaining parameters $a_0 = 0.002$, $\alpha_0 = 0.01$, $\rho_c = \rho_0/2 = 0.5$, $G = 2$, $D_\rho = 0.04$, $K = 0.01$.

## Approximation for growth-induced alignment

The following approximation and equations are received from *Dell'Arciprete et al., 2018*. These approximations were used to explain the tangential alignment of bacteria at the edge of growing colonies. The equation of motion for 2D nematodynamics without any free energy and no spatial variation of $Q$ can be written as:

$$\frac{\partial Q_{\alpha\beta}}{\partial t} + v_\gamma \partial_\gamma Q_{\alpha\beta} = \xi u_{\alpha\beta} - Q_{\alpha\gamma}\omega_{\gamma\beta} + \omega_{\alpha\gamma}Q_{\gamma\beta} \tag{17}$$

If we assume

$$v_r = g\left(r\right)r \tag{18}$$

From this formulation we can conclude that:

$$v_\alpha = g\left(r\right)x_\alpha \tag{19}$$

where $x_\alpha$ is the Cartesian component of the position vector

$$\partial_\beta v_\alpha = g\delta_{\alpha\beta} + g'rr_\alpha r_\beta \tag{20}$$

where $r_\alpha = \frac{x_\alpha}{r}$. Now this tensor is symmetric, with calculating $u_{\alpha\beta}$ and $\omega_{\alpha\beta} = 0$ putting it in *Equation 17* above we get:

$$\frac{\partial Q_{\alpha\beta}}{\partial t} = \xi g'r\left(r_\alpha r_\beta - \frac{\delta_{\alpha\beta}}{2}\right) \tag{21}$$

With writing $Q_{\alpha\beta}$ using:

$$Q_{\alpha\beta} = S\begin{bmatrix} \cos^2\theta - \frac{1}{2} & \sin\theta\cos\theta \\ \sin\theta\cos\theta & \sin^2\theta - \frac{1}{2} \end{bmatrix} \tag{22}$$

$$= \frac{S}{2}\begin{bmatrix} \cos(2\theta) & \sin(2\theta) \\ \sin(2\theta) & -\cos(2\theta) \end{bmatrix} \tag{23}$$

In polar coordinates $(r, \phi)$ right-hand side of *Equation 21* is:

$$\frac{\partial Q}{\partial t} = \frac{\xi g'r}{2}\begin{bmatrix} \cos\left(2\phi\right) & \sin\left(2\phi\right) \\ \sin\left(2\phi\right) & -\cos\left(2\phi\right) \end{bmatrix} \tag{24}$$

If we combine *Equation 23* and *Equation 24*:

$$-S\sin\left(2\theta\right)\frac{d\theta}{dt} = \frac{\xi g'r}{2}\cos\left(2\phi\right) \tag{25}$$

$$S\cos\left(2\theta\right)\frac{d\theta}{dt} = \frac{\xi g'r}{2}\sin\left(2\phi\right) \tag{26}$$

Multiply first equation (*Equation 25*) by $-sin\left(2\phi\right)$ and second equation (*Equation 26*) by $cos\left(2\phi\right)$ and sum them up:

$$\frac{d\theta}{dt} = \frac{\xi g'r}{2S}\left[\sin\left(2\phi\right)\cos\left(2\theta\right) - \cos\left(2\phi\right)\sin\left(2\theta\right)\right] \tag{27}$$

$$\frac{d\theta}{dt} = \frac{\xi g'r}{2S}\sin\left(2\left(\phi - \theta\right)\right) \tag{28}$$

Thus if $g' > 0$, equation above has a stable equilibrium for $\theta = \phi$ (aster).

## Code availability

The codes utilized previously published open-source software from https://depts.washington.edu/soslab/gro/ and are made available on GitHub (https://github.com/mustafa-basaran/Large_Scale_Orientation_Bacteria, swh:1:rev:fd7673254fa57874676b183f9a944d9e457c3ac0; *Basaran, 2021*).

## Acknowledgements

This work was supported by an EMBO installation Grant (IG 3275, AK) and BAGEP young investigator award (AK). We thank Sharad Ramanathan for suggestions about bacterial competitions. We thank

Julia Yeomans for discussions and suggestions. We thank FM Ramazanoğlu, A Kabakçıoğlu, and M Muradoğlu for critical reading of the manuscript.

## Additional information

### Funding

| Funder | Grant reference number | Author |
|--------|------------------------|--------|
| EMBO | Installation Grant 3275 | Askin Kocabas |
| BAGEP | Young investigator award | Askin Kocabas |

The funders had no role in study design, data collection and interpretation, or the decision to submit the work for publication.

### Author contributions

Mustafa Basaran, Conceptualization, Data curation, Formal analysis, Investigation, Methodology, Software, Validation, Visualization, Writing - original draft, Writing - review and editing; Y Ilker Yaman, Conceptualization, Formal analysis, Investigation; Tevfik Can Yüce, Software, Visualization; Roman Vetter, Methodology, Resources, Software, Visualization; Askin Kocabas, Conceptualization, Data curation, Formal analysis, Funding acquisition, Investigation, Methodology, Project administration, Resources, Software, Supervision, Validation, Visualization, Writing - original draft, Writing - review and editing

### Author ORCIDs

Mustafa Basaran http://orcid.org/0000-0002-1895-254X
Y Ilker Yaman http://orcid.org/0000-0003-4094-616X
Tevfik Can Yüce http://orcid.org/0000-0002-6888-2690
Roman Vetter http://orcid.org/0000-0003-2901-7036
Askin Kocabas http://orcid.org/0000-0002-6930-1202

### Decision letter and Author response

Decision letter https://doi.org/10.7554/eLife.72187.sa1
Author response https://doi.org/10.7554/eLife.72187.sa2

## Additional files

### Supplementary files
• Transparent reporting form

### Data availability

The critical experimental data generated or analyzed during this study are provided as supporting video files. Code Availability: The codes utilized previously published open-source software from https://depts.washington.edu/soslab/gro/ and are made available on GitHub (https://github.com/mustafa-basaran/Large_Scale_Orientation_Bacteria, copy archived at swh:1:rev:fd7673254fa57874676b183f9a944d9e457c3ac0).

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
