## [Editor Report]

The growth of bacterial colonies on solid substrates is a common assay used in a variety of settings, from probing bacterial organization in biofilms to spatial population genetics. The common setup is an outward growing colony from a central seed. In this work, Basaran et al., study a colony growing inward from an annulus. The authors show that this geometrical modification has profound consequences on the alignment of rod-shaped bacteria. This is caused by a flow alignment effect, and lead to a radial ordering reminiscent of an aster or +1 topological defect. This result is motivated by experimental observations with *E. coli* and interpreted using modern active matter theories, with ample support from extensive numerical simulations of detailed finite element and continuum models.

---

## [Decision Letter]

**Decision letter after peer review:**

Thank you for submitting your article "Large-scale orientational order in bacterial colonies during inward growth" for consideration by *eLife*. Your article has been reviewed by 2 peer reviewers, and the evaluation has been overseen by a Reviewing Editor and Aleksandra Walczak as the Senior Editor. The following individual involved in review of your submission has agreed to reveal their identity: Suraj Shankar (Reviewer #2).

Essential revisions:

1) Substantiate your claim that the ordering is caused by flow alignment effect by reporting the flow distribution observed experimentally and comparing it to the theoretical prediction (reviewer #2 question 1)

2) Specify how stress is computed in the simulations, and discuss the extent to which it can be determined experimentally. (reviewer #1 question 2 and reviewer #1 question 1))

3) Improve the discussion of the multilayering transition, and in particular the role of in-plane stress in the process along the lines suggested by reviewer #1 question 1 and reviewer #2 question 2 and question 4

4) Provide a point-by-point answer to all the reviewers comments and questions.

*Reviewer #1:*

This manuscript combines experiments, theory and simulation to study bacterial patterns in a colony growing inwards, from an annulus. The authors find that in this geometry growth leads to leads to the formation of an aster, which is a defect of topological charge +1, where bacteria tend to align in the radial direction. Previously, a growing colony of bacteria was reported to lead to nematic microdomain formation, with bacteria aligning tangentially at the colony edge, with half-integer defects in between microdomains. Overall I think this work is a nice example of an application of topology to bacterial biophysics, and is likely to appeal the growing active matter community.

The combination of experiments, theory and simulation renders the results convincing. It is nice that the theory allows to get a mechanistic and fundamental understanding of the reason for the aster formation, which can be traced back to the nonlinearity of the bacterial velocity/flow profiles. It is also nice that the simulations reproduce all previous observations in different geometries, providing validation of the current results.

Although overall the methodology is sound and the results appear robust, some clarifications are sought as follows.

1. In the orientational patterns in Figure 3 it appears that some bacteria align out of the plane. If this is the case, and not a visualisation issue, it would be good to mention the relevance of the verticalisation, or to perform simulations where this is disallowed as in a growing monolayer.

2. It would be good to describe in a bit more detail how the plotted stress is computed in the simulation and how it could be estimated experimentally.

3. The patterns in multilayer colonies are of interest but it would be good to add a discussion of the reason why different surfaces lead to results which are so different.

Recommendations for the authors:

As mentioned I find this work interesting and I believe it will stimulate discussion in the active matter and bacterial biophysics community. Specifying more in detail what said in the public summary:

1. I would encourage authors to discuss the issue of verticalisation and to specify more clearly where the director can escape to the 3rd dimensions in their monolayer colony simulations (I understand this is possible in multilayer colony simulations).

2. Please specify how stress is computed, is this a component or a scalar built starting from the stress tensor? This should be specified in the main text.

3. It could be said more clearly that the inward colony is a particular geometry where g' (in the manuscript notation) is non-zero, there could be others as well. Simulations of expanding monolayers I believe suggested a possible radial orientation at the edge for sufficiently large colonies, I think this is mentioned in Ref. 42.

4. Is the drop in nematic order e.g. in Figure 3d, which occurs close to the critical radius for which v is about 0, robust and can be understood via flow-mediated rotation as well? Sorry if I missed this.

5. The final part on competition is interesting but I am missing the difference between e.g. a and c in Figure 6 or in general where red and green are swapped but the ratio between long and short is the same. Are these not symmetric?

6. A very minor point: why is the left-hand side of Eq. 6 not written as a material derivative?

*Reviewer #2:*

The growth of bacterial colonies on solid substrates is a common assay used in a variety of settings, from probing bacterial organization in biofilms to spatial population genetics. In this work, Basaran et al., study how crowded bacterial colonies invade an enclosed space, in contrast to the more common setup of a growing monolayer that expands outward in a unconstrained fashion. This seemingly innocuous modification has dramatic consequences as the authors show. Geometric confinement and growth from cell division dictate a characteristic velocity field that vanishes at a finite radius and the resulting shear flow aligns bacteria to orient in a radial fashion. The colony wide radial ordering is reminiscent of an aster or +1 topological defect seen in liquid crystals. A key point emphasized in the paper is that such large scale ordering of bacteria does not occur in outward expanding colonies, but is typical of inward growth. This result is motivated by experimental observations with *E. coli* and interpreted using modern active matter theories, with ample support from extensive numerical simulations of detailed finite element and continuum models. The structure and flow generated by radially oriented bacteria is shown to affect multilayering, both in simulations and in experiments with prepatterned annular rings of bacteria. Finally the authors demonstrate a potential biological significance of such orientational order by considering (in silico) two competing bacterial strains that are genetically neutral but have different lengths. The enhanced propensity of the longer bacterium to radially order endows it with a selective advantage to out compete the shorter strain in a spatial setting.

I find that most of the claims are well substantiated and justified by the data presented, though a few points need better support. The main strength of the paper is the involved and detailed numerical modelling employed to describe the invasion of bacterial colonies. It is an impressive amount of computational work. While some of the main points such as the emergence of a radial aster accompanied by a sign changing velocity field in inward growth are recapitulated in experimental data, the authors only make qualitative comparisons with the model. This I feel is a missed opportunity that can be easily remedied given the present data, particularly in the case of patterned colonies (Figure 5). For instance, it is unclear what selects the critical radius Rc, and how it is determined by the initial inoculation geometry. A more quantitative comparison between the experimental and numerical data might help elucidate this point more.

Another weakness is in the discussion surrounding multilayer formation which is a bit disjointed and separate from the first part of the paper. The primary claim rests on a plausible argument suggesting compressive stresses near the critical radius cause buckling and multilayer formation, but the current data is only partially convincing. Figure 4 only demonstrates the presence of multilayering in the finite element simulations and in the experiments but does not validate the suggested mechanism. A straightforward resolution would be to present measurements (at least from the simulation) of the hoop and radial stress in the monolayer and correlate it with the flow, radial order and buckling. The experimental demonstrations also lack descriptions of simple details regarding their setup at various places, which needs to be improved.

In the last section on bacterial competition with differential length strains, I feel the claim regarding the enhanced radial order in the longer bacterium is also not sufficiently substantiated. The red and green curves in Figure 6a-f are meant to demonstrate this claim, but all three plots look rather similar and it is unclear how statistically significant the difference between the curves really is. Either the data must be presented along with a statistical analysis demonstrating significant difference in radial order or the claim must be toned down. Note, the statement about enhanced radial order doesn't necessarily affect (though it is suggested as a causal mechanism) the more significant and consequential result regarding the excess area fraction of the longer bacterium over the shorter, which does demonstrate the claimed selective advantage.

Recommendations for the authors:

Most of my comments are regarding presentation. The current structure of the paper is a bit confusing with the many different numerical simulation methods and experimental setups used. The following suggestions may help clarify and improve the paper. I recommend publication once the comments and questions are satisfactorily answered.

1) The experimental data in Figure 1 only shows the presence of radial order (Figure 1d), but the following figure panels (the rest of Figure 1, and also Figures 2-3) provide details of both radial ordering and the velocity profile. The latter is shown to be the more important and primary ingredient (in numerical simulations) that causes radial alignment of bacteria. In this regard the experimental results of Figure 1only provides circumstantial evidence for the proposed flow induced mechanism. Only much later, in the section on "Inward growing domains in monolayer colonies", is it mentioned that "So far, we studied naturally emerged inward growing domains on agar surfaces. These domains are randomly formed across the plate. In these experiments particularly, the confinement is defined by the crowded multilayered environment. Thus, observing critical radius and detailed velocity profiles are not possible." The lithographically patterned bacterial rings on a PC surface shown in Figure 5 do allow the measurement of both radial order and the flow field, both of which display the qualitative features predicted by the model.

I would hence suggest moving part of the results on patterned bacteria to the beginning, as it provides a clearer and more striking comparison with the numerical work. It should also be clarified when the experiments in Figure 1 are discussed whether the bacterial colony exhibits multilayering outside the field of view shown, in which case prevents a good measurement of the flow field is not possible. Several details about Figure 5 are missing though. What bacteria were used for these experiments and what are the time stamps for the various snapshots in panels a and b? Are the plots in d and e also averaged over 4 independent experiments? Presumably, the whiter regions in the phase contrast images are small patches of double or triple layers. These points should be clarified and explicitly mentioned.

2) Is it understood why decreasing substrate friction permits larger monolayers? Is there a way to test this using the 3D FEM simulation? It would be helpful if some intuition could be given about the connection between friction and buckling, perhaps as a balance of traction forces and growth pressure? On line 256, it is mentioned that "Previous studies showed that, particularly, the surface friction and stress accumulation are responsible for the formation of this multi-layered structures". What previous studies are being referred to here?

For the FEM simulations modelling biofilm forming bacteria, what parameters were modified? The methods section primarily doesn't seem to provide any details about this.

3) While one of the main strengths of the paper is the sophisticated modelling, the use of so many different methods (2D and 3D FEM, continuum theory) makes the reader wonder what benefit is gained from one method over the other. In particular, it is unclear what specific role the continuum active nematic model plays in understanding the phenomena of radial ordering in growing bacterial collectives. All of the results from the continuum model in Figure 3 are essentially the same as obtained from the detailed FEM simulation in Figure 2. Although the use of active nematic continuum models to describe growing bacteria has become fashionable in recent years, it is unclear in the current paper if it offers any new insight that may not be gained otherwise.

I would suggest that the authors reword this section to highlight specific benefits and insights gained from using the continuum model, or move some of the discussion to the SI. It might also help to frame the benefit of the continuum model as generalizing the results (by virtue of coarse-graining over irrelevant microscopic details) beyond the specific bacterial and particle based implementation. As of now, the relevance of the active nematic model is not apparent.

4) I don't quite understand the line "Herein, experimentally we only observed inward growing domains around the inner edge because the accumulated stress triggers multi-layer formation" (Line 206-207). What does stress triggered multilayer formation have to do with observing (or not) inward growth of the colony? I thought the former was causal consequence of the latter and not the other way around.

As mentioned earlier, I think it would be useful to present stress profiles (both radial and hoop components separately as a function of r) from the simulations to substantiate the mechanism triggering buckling and multilayer formation. I suspect that the annular geometry and self-induced confinement due to cell proliferation generates a compressive hoop stress that underlies escape into the third dimension. In this regard it might be worth making a comparison with a classic wrinkling instability of thin sheets in the so-called Lame problem (see for instance, Davidovitch et al. "Prototypical model for tensional wrinkling in thin sheets." PNAS 108.45 (2011): 18227-18232.) In the passive elastic case, outward tension combined with the annular geometry of the elastic film generates compressive hoop stresses that are resolved by wrinkling. Analogously, growth (instead of outward external tension) combined with the geometry leads to a potentially similar effect, now in an active bacterial layer, which resolves the stresses by forming multiple layers, rather than wrinkling.

---

## [Author Response]

Essential revisions:1) Substantiate your claim that the ordering is caused by flow alignment effect by reporting the flow distribution observed experimentally and comparing it to the theoretical prediction (reviewer #2 question 1)

We have experimentally measured the velocity of the flow in Figure 1 d.

2) Specify how stress is computed in the simulations, and discuss the extent to which it can be determined experimentally. (reviewer #1 question 2 and reviewer #1 question 1))

We have provided detailed information about stress calculations in the methods section. We also discuss how the stress profile can be measured. See section “Calculating stress distribution in a growing colony”

3) Improve the discussion of the multilayering transition, and in particular the role of in-plane stress in the process along the lines suggested by reviewer #1 question 1 and reviewer #2 question 2 and question 4

We have provided new FEM simulation results to clarify the multi-layering process. We have also added a new supplementary figure to the manuscript.

Reviewer #1:This manuscript combines experiments, theory and simulation to study bacterial patterns in a colony growing inwards, from an annulus. The authors find that in this geometry growth leads to leads to the formation of an aster, which is a defect of topological charge +1, where bacteria tend to align in the radial direction. Previously, a growing colony of bacteria was reported to lead to nematic microdomain formation, with bacteria aligning tangentially at the colony edge, with half-integer defects in between microdomains. Overall I think this work is a nice example of an application of topology to bacterial biophysics, and is likely to appeal the growing active matter community.The combination of experiments, theory and simulation renders the results convincing. It is nice that the theory allows to get a mechanistic and fundamental understanding of the reason for the aster formation, which can be traced back to the nonlinearity of the bacterial velocity/flow profiles. It is also nice that the simulations reproduce all previous observations in different geometries, providing validation of the current results.Although overall the methodology is sound and the results appear robust, some clarifications are sought as follows.1. In the orientational patterns in Figure 3 it appears that some bacteria align out of the plane. If this is the case, and not a visualisation issue, it would be good to mention the relevance of the verticalisation, or to perform simulations where this is disallowed as in a growing monolayer.

We thank the reviewer for highlighting this issue. In our simulations, we have vertical force (representing surface tension) keeping the bacteria on the surface. During multilayer formation bacteria transiently orient out of the plane and then become horizontal again in the second layer. We agree that all these issues are strongly related to verticalization events that stochastically trigger multilayering. In the current version, we have emphasized the relation between verticalization and multilayer formation. The following part of the manuscript has been modified.

“We observed that on a regular agar surface, again multi-layer formation dominates the overall colony morphology. Only very narrow monolayer regions are observable around the inner and outer edges of the colony. We then focus our attention on how to eliminate this multilayering process. A simple glass or PDMS confinement cannot eliminate this multilayering (Figure 4—figure supplement 1). Previous studies showed that attractive biochemical interactions between bacteria and surface could generate additional strong friction force [41]. Altogether friction force, stress accumulation, and verticalization of bacteria in a monolayer colony trigger the formation of these multi-layered structures. This process is mainly controlled by the competition between vertical force and lateral compression in the colony [1, 13, 41, 42]. Above the critical stress level, the orientation of rod-shaped bacteria becomes unstable and triggers the extrusion. Performing FEM simulations, we noticed that this extrusion process occurs around the center of the annuls and it can be controlled by surface friction (Figure 5—figure supplement 2). Although we don’t know the detailed biological mechanism behind the friction force it is evident that minimizing the surface friction can increase the size of the monolayer colony. Then, we tested the same bacterial patterning on different membranes to find a surface with low friction by minimizing biochemical interaction. We noted that only polycarbonate (PC) surfaces are useful for this purpose, and they support large monolayer colonies while providing a sufficient bacterial growth rate (Figure 5—figure supplement 3, Figure 5-video 1, see Materials and methods).”

2. It would be good to describe in a bit more detail how the plotted stress is computed in the simulation and how it could be estimated experimentally.

We have added the details of stress calculation to the method section and the other stress components are also given in Figure 2—figure supplement 1.

3. The patterns in multilayer colonies are of interest but it would be good to add a discussion of the reason why different surfaces lead to results which are so different.

We fully agree with the referee that the substrate has a significant impact on the colony size. We believe this is originating from the additional surface friction mechanism controlled by biochemical interactions. Duvarney et.al. (41) recently highlighted this issue. Our results also support this claim. We have discussed this contribution more clearly and we also provided additional FEM simulations to support our experimental results.

Recommendations for the authors:As mentioned I find this work interesting and I believe it will stimulate discussion in the active matter and bacterial biophysics community. Specifying more in detail what said in the public summary:1. I would encourage authors to discuss the issue of verticalisation and to specify more clearly where the director can escape to the 3rd dimensions in their monolayer colony simulations (I understand this is possible in multilayer colony simulations).

We have improved the discussion of verticalization and provided additional FEM simulations (Figure 4—figure supplement 2) to support our claims.

2. Please specify how stress is computed, is this a component or a scalar built starting from the stress tensor? This should be specified in the main text.

We have added the details to the methods section. We also provided the Fiugre 2—figure supplement 1 to show other stress components in 2D simulations.

3. It could be said more clearly that the inward colony is a particular geometry where g' (in the manuscript notation) is non-zero, there could be others as well. Simulations of expanding monolayers I believe suggested a possible radial orientation at the edge for sufficiently large colonies, I think this is mentioned in Ref. 42.

We thank the referee for pointing out this issue. Ref 42 (Ref 45 in the revised version) reports interesting radial alignment around the leading edge of colonies. We don’t think this is triggered by the size of the colonies. This alignment might be a local effect where the radial symmetry is disappeared. We also think that this alignment could also result from the biofilm formation process. Unlike *B. subtilis, some E. coli* strains may generate strong adhesive proteins (Ag43) which can keep the bacteria attached together with extra cellular matrix. This biofilm structure could lead additional growth profile and compressibility. We also recently observed this effect in our experiments. After gap closure some *E. coli* strains could maintain the radial alignment, however, in *B. subtilis* radial alignment converges to random microdomain formations. Our observations still not fully conclusive. We would like to leave these details to a future work.

4. Is the drop in nematic order e.g. in Figure 3d, which occurs close to the critical radius for which v is about 0, robust and can be understood via flow-mediated rotation as well? Sorry if I missed this.

This observation is robust. However, we don’t exactly know why we have this significant drop. We used Comsol FEM simulation platform to solve the coupled differential equations. Still, discretization , triangulation and numerical error could be responsible for.

5. The final part on competition is interesting but I am missing the difference between e.g. a and c in Figure 6 or in general where red and green are swapped but the ratio between long and short is the same. Are these not symmetric?

We thank the referee for highlighting this confusing part. The difference was not visible due to the large range of the order parameter. We have provided the zoomed version to highlight the difference. This scenario is the most challenging situation to test the competition. In order to eliminate the segmentation effect, we tried to start the competition with a highly packed and random distribution. Up to a specific annulus shape, both red and green bacteria have the same division length, and they can form a random distribution. This initial random profile is given in Figure 6—figure supplement 1. After this stage, we changed the division length. The ratio between red and green bacteria may have some bias toward a specific type. However, with the same annulus initial shape, switching the division lengths significantly changes the final bacterial distribution and packing fraction around the leading edge which is also given in Figure 6 g. We have clarified this point in the manuscript.

6. A very minor point: why is the left-hand side of Eq. 6 not written as a material derivative?

We have fixed this issue.

Reviewer #2:The growth of bacterial colonies on solid substrates is a common assay used in a variety of settings, from probing bacterial organization in biofilms to spatial population genetics. In this work, Basaran et al., study how crowded bacterial colonies invade an enclosed space, in contrast to the more common setup of a growing monolayer that expands outward in a unconstrained fashion. This seemingly innocuous modification has dramatic consequences as the authors show. Geometric confinement and growth from cell division dictate a characteristic velocity field that vanishes at a finite radius and the resulting shear flow aligns bacteria to orient in a radial fashion. The colony wide radial ordering is reminiscent of an aster or +1 topological defect seen in liquid crystals. A key point emphasized in the paper is that such large scale ordering of bacteria does not occur in outward expanding colonies, but is typical of inward growth. This result is motivated by experimental observations with *E. coli* and interpreted using modern active matter theories, with ample support from extensive numerical simulations of detailed finite element and continuum models. The structure and flow generated by radially oriented bacteria is shown to affect multilayering, both in simulations and in experiments with prepatterned annular rings of bacteria. Finally the authors demonstrate a potential biological significance of such orientational order by considering (in silico) two competing bacterial strains that are genetically neutral but have different lengths. The enhanced propensity of the longer bacterium to radially order endows it with a selective advantage to out compete the shorter strain in a spatial setting.I find that most of the claims are well substantiated and justified by the data presented, though a few points need better support. The main strength of the paper is the involved and detailed numerical modelling employed to describe the invasion of bacterial colonies. It is an impressive amount of computational work. While some of the main points such as the emergence of a radial aster accompanied by a sign changing velocity field in inward growth are recapitulated in experimental data, the authors only make qualitative comparisons with the model. This I feel is a missed opportunity that can be easily remedied given the present data, particularly in the case of patterned colonies (Figure 5). For instance, it is unclear what selects the critical radius Rc, and how it is determined by the initial inoculation geometry. A more quantitative comparison between the experimental and numerical data might help elucidate this point more.

We fully agree with the referee that we couldn’t calculate the position of the critical radius analytically. However, our experimental results are indicating that it does not show any interesting trend. The position looks stable around the center of the annulus shape and is simply defined by the inoculation geometry. It is also possible that the asymmetry of the velocity profile can change the position of the critical radius, particularly at later stage. However, the stochastic nature of cell divisions (both in experiments and simulations) does not allow observing this small shift. In our new FEM simulation (Figure5 – —figure supplement 2) we plotted the velocity profile and the stress distribution. Similarly, the variation is actually very large. We also observed the first 5 extrusion events of the bacteria is stochastic. At later stage multilayering process completely changes entire dynamics. We believe it is almost impossible to quantitatively compare critical radius at this stage.

Another weakness is in the discussion surrounding multilayer formation which is a bit disjointed and separate from the first part of the paper. The primary claim rests on a plausible argument suggesting compressive stresses near the critical radius cause buckling and multilayer formation, but the current data is only partially convincing. Figure 4 only demonstrates the presence of multilayering in the finite element simulations and in the experiments but does not validate the suggested mechanism. A straightforward resolution would be to present measurements (at least from the simulation) of the hoop and radial stress in the monolayer and correlate it with the flow, radial order and buckling. The experimental demonstrations also lack descriptions of simple details regarding their setup at various places, which needs to be improved.

We have provided additional FEM results in Figure5 – —figure supplement 2(shown in the essential revision part). An essential part of the multilayering process has been investigated in the literature *[1, 13, 41, 42]*. We have also provided additional references *[41]* to clarify this point.

In the last section on bacterial competition with differential length strains, I feel the claim regarding the enhanced radial order in the longer bacterium is also not sufficiently substantiated. The red and green curves in Figure 6a-f are meant to demonstrate this claim, but all three plots look rather similar and it is unclear how statistically significant the difference between the curves really is. Either the data must be presented along with a statistical analysis demonstrating significant difference in radial order or the claim must be toned down. Note, the statement about enhanced radial order doesn't necessarily affect (though it is suggested as a causal mechanism) the more significant and consequential result regarding the excess area fraction of the longer bacterium over the shorter, which does demonstrate the claimed selective advantage.

We thank the referee for highlighting this confusing part. We kindly don’t agree to the referee at this point. We believe the difference is significant, but it was not visible in the original figures due to the large range of order parameters. We have provided the zoomed version to highlight this difference. The original graphs are given in the supplementary figure.

Recommendations for the authors:Most of my comments are regarding presentation. The current structure of the paper is a bit confusing with the many different numerical simulation methods and experimental setups used. The following suggestions may help clarify and improve the paper. I recommend publication once the comments and questions are satisfactorily answered.1) The experimental data in Figure 1 only shows the presence of radial order (Figure 1d), but the following figure panels (the rest of Figure 1, and also Figures 2-3) provide details of both radial ordering and the velocity profile. The latter is shown to be the more important and primary ingredient (in numerical simulations) that causes radial alignment of bacteria. In this regard the experimental results of Figure 1only provides circumstantial evidence for the proposed flow induced mechanism. Only much later, in the section on "Inward growing domains in monolayer colonies", is it mentioned that "So far, we studied naturally emerged inward growing domains on agar surfaces. These domains are randomly formed across the plate. In these experiments particularly, the confinement is defined by the crowded multilayered environment. Thus, observing critical radius and detailed velocity profiles are not possible." The lithographically patterned bacterial rings on a PC surface shown in Figure 5 do allow the measurement of both radial order and the flow field, both of which display the qualitative features predicted by the model.I would hence suggest moving part of the results on patterned bacteria to the beginning, as it provides a clearer and more striking comparison with the numerical work. It should also be clarified when the experiments in Figure 1 are discussed whether the bacterial colony exhibits multilayering outside the field of view shown, in which case prevents a good measurement of the flow field is not possible.

Following the reviewers’ suggestions, we have provided an experimental velocity profile for the inward colony used in Figure 1. Figure 2—figure supplement 4 also provides some experimental results. On randomly seeded plates, merging colonies form very dense and crowded regions. Around these regions, we cannot image individual bacteria and it is not possible to quantify their detailed dynamics. However, this system is more accessible to everyone, so it is more important to study first. With this sentence, we simply tried to emphasize the challenges and limitations of this system. Furthermore, this revised manuscript will be the 4^th^ version posted online and we would like to follow the same logic instead of moving the membrane experiments.

Several details about Figure 5 are missing though. What bacteria were used for these experiments and what are the time stamps for the various snapshots in panels a and b? Are the plots in d and e also averaged over 4 independent experiments? Presumably, the whiter regions in the phase contrast images are small patches of double or triple layers. These points should be clarified and explicitly mentioned.

We have provided more information and clearly indicated the bacterial strain in each figure.

2) Is it understood why decreasing substrate friction permits larger monolayers? Is there a way to test this using the 3D FEM simulation? It would be helpful if some intuition could be given about the connection between friction and buckling, perhaps as a balance of traction forces and growth pressure? On line 256, it is mentioned that "Previous studies showed that, particularly, the surface friction and stress accumulation are responsible for the formation of this multi-layered structures". What previous studies are being referred to here?

We thank the reviewers for bringing up this point. This process is mainly controlled by the competition between vertical forces and lateral compression in the colony. Above the threshold it becomes unstable. We have modified this part in the manuscript to highlight these issues. We have demonstrated the same process using 3D FEM simulations. We also provided additional references indicating the importance of all the important factors.

“We observed that on a regular agar surface, again multi-layer formation dominates the overall colony morphology. Only very narrow monolayer regions are observable around the inner and outer edges of the colony. We then focus our attention on how to eliminate this multilayering process. A simple glass or PDMS confinement cannot eliminate this multilayering (Figure 4—figure supplement 1). Previous studies showed that attractive biochemical interactions between bacteria and surface could generate additional strong friction force [41]. Altogether friction force, stress accumulation, and verticalization of bacteria in a monolayer colony trigger the formation of these multi-layered structures. This process is mainly controlled by the competition between vertical force and lateral compression in the colony [1, 13, 41, 42]. Above the critical stress level, the orientation of rod-shaped bacteria becomes unstable and triggers the extrusion. Performing FEM simulations, we noticed that this extrusion process occurs around the center of the annuls and it can be controlled by surface friction (Figure 5—figure supplement 2). Although we don’t know the detailed biological mechanism behind the friction force it is evident that minimizing the surface friction can increase the size of the monolayer colony. Then, we tested the same bacterial patterning on different membranes to find a surface with low friction by minimizing biochemical interaction. We noted that only polycarbonate (PC) surfaces are useful for this purpose, and they support large monolayer colonies while providing a sufficient bacterial growth rate (Figure 5—figure supplement 3, Figure 5-video 1, see Materials and methods).”

For the FEM simulations modelling biofilm forming bacteria, what parameters were modified? The methods section primarily doesn't seem to provide any details about this.

We have provided the details in the method section.

3) While one of the main strengths of the paper is the sophisticated modelling, the use of so many different methods (2D and 3D FEM, continuum theory) makes the reader wonder what benefit is gained from one method over the other.

We agree that using different simulation platforms in the same paper might be confusing. However, it is inevitable to use all these tools to clarify different aspects of the system. GRO provides a fast simulation platform in 2D, and FEM is capable of handling 3D complex environments. We have emphasized all these limitations and advantages in the manuscript.

In particular, it is unclear what specific role the continuum active nematic model plays in understanding the phenomena of radial ordering in growing bacterial collectives. All of the results from the continuum model in Figure 3 are essentially the same as obtained from the detailed FEM simulation in Figure 2. Although the use of active nematic continuum models to describe growing bacteria has become fashionable in recent years, it is unclear in the current paper if it offers any new insight that may not be gained otherwise.I would suggest that the authors reword this section to highlight specific benefits and insights gained from using the continuum model, or move some of the discussion to the SI. It might also help to frame the benefit of the continuum model as generalizing the results (by virtue of coarse-graining over irrelevant microscopic details) beyond the specific bacterial and particle based implementation. As of now, the relevance of the active nematic model is not apparent.

Following the reviewers’ suggestions, we have reworded this section to highlight this issue. Actually, the theory is not our main expertise. We particularly developed this section based on the feedback we received from theorists working in this field. We aimed to identify if there are any steric effects specific to bacterial colonies. Our continuum results are showing that this is a generic response. We have highlighted all these issues in the manuscript.

4) I don't quite understand the line "Herein, experimentally we only observed inward growing domains around the inner edge because the accumulated stress triggers multi-layer formation" (Line 206-207). What does stress triggered multilayer formation have to do with observing (or not) inward growth of the colony? I thought the former was causal consequence of the latter and not the other way around.

Here we tried to highlight the limitation of this approach resulting from the multilayering process. We have clarified this sentence.

“Herein, experimentally we observed inward growing domains only around the inner edge surrounded by dense multi-layered structures. This is because merging colonies and the accumulated stress trigger multi-layer formation and limit the size of the monolayer region around the edge (Figure 4—figure supplement 1).”

As mentioned earlier, I think it would be useful to present stress profiles (both radial and hoop components separately as a function of r) from the simulations to substantiate the mechanism triggering buckling and multilayer formation. I suspect that the annular geometry and self-induced confinement due to cell proliferation generates a compressive hoop stress that underlies escape into the third dimension. In this regard it might be worth making a comparison with a classic wrinkling instability of thin sheets in the so-called Lame problem (see for instance, Davidovitch et al. "Prototypical model for tensional wrinkling in thin sheets." PNAS 108.45 (2011): 18227-18232.) In the passive elastic case, outward tension combined with the annular geometry of the elastic film generates compressive hoop stresses that are resolved by wrinkling. Analogously, growth (instead of outward external tension) combined with the geometry leads to a potentially similar effect, now in an active bacterial layer, which resolves the stresses by forming multiple layers, rather than wrinkling.

We would like to thank the reviewer for pointing out this interesting point. This Lame problem was actually our initial starting point. However, later we noticed that all the details are completely different. To clarify these differences, we have provided radial and hoop stress profiles and we added the following text to the manuscript.

“We should emphasize that inward growing bacterial colonies and wrinkling thin circular sheets have geometric similarities [54]. In these elastic circular objects, under axisymmetric tensile load, azimuthal stress (hoop stress, σ_θθ) show transition from tensile to compressive profile which eventually creates radial wrinkling pattern below critical radius. However, unlike elastic objects, growing bacterial colonies can only develop compressive stress due to negligible attractive force between bacteria.”